# Preferences, trust, and performance in youth business groups

Stein T. Holden [1]*, Mesfin Tilahun [1,2]*

1 School of Economics and Business, Norwegian University of Life Sciences, Ås, Norway, 2 Department of Economics, Mekelle University, Mekelle, Ethiopia

☯ These authors contributed equally to this work.
* stein.holden@nmbu.no (STH); mesfin.tilahun.gelaye@nmbu.no, mesfintilahungelaye@gmail.com (MT)

**Data Availability Statement:** The minimal underlying dataset and the econometric estimation methods are in STATA files as Supporting Information files (See S1 Dataset of youth groups, S2 Dataset of youth group members, and S1 Do-file for S1 and S2 Datasets).

## Abstract

We study how social preferences and norms of reciprocity are related to generalized (outgroup) and particularized (ingroup) trust among members of youth business groups in northern Ethiopia. The Ethiopian government promotes youth employment among land-poor rural youth by allocating them rehabilitated communal lands for the formation of sustainable businesses. The typical sustainable production activities that the groups can invest in include apiculture, forestry, horticulture, and livestock production. Our study used incentivized experiments to elicit social preferences, trust, and trustworthiness. We use data from 2427 group members in 246 functioning business groups collected in 2019. Altruistic and egalitarian preferences were associated with stronger norms to reciprocate, higher outgroup and ingroup trustworthiness and trust while spiteful and selfish preferences had opposite effects. The social preferences had both direct and indirect effects (through the norm to reciprocate) on trustworthiness and trust. Ingroup trust was positively correlated with a number of group performance indicators.

## Introduction

The pressures on the natural resource base from population growth, economic development, and climate change are increasing and making it harder for people to carve out sustainable livelihoods within vulnerable agro-ecologies. Such pressures are particularly increasing in parts of Sub-Saharan Africa where population densities are high, and rainfall is limited and variable [1]. The Ethiopian highlands is one such "environmental hot-spot" where rural transformation is needed to meet the needs of the new generation in search of new livelihood opportunities as they cannot only continue in the footpaths of their parents [2]. The shrinking farm sizes have now reached a level that implies that further splitting of farms among the children leads to micro-farms that require complementary sources of income for those having such farms. One policy initiative in northern Ethiopia has been to allocate rehabilitated communal lands to groups of landless and land-poor youth that aim to establish a livelihood in their rural home community. Many may question whether youth are able to organize themselves and jointly manage a common-pool resource in a sustainable way [3–5]. It is therefore

**Funding:** Stein T. Holden Grant Number: 288238
The Research Council of Norway https://www.
forskningsradet.no/en/ The funders had no role in
study design, data collection and analysis, decision
to publish, or preparation of the manuscript.

**Competing interests:** The authors have declared
that no competing intrests exist.

both a bold and a risky policy initiative that we are studying. There exist very few large-N studies of "kick-started" natural resource management livelihood groups like this [6].

A census of more than 700 such groups by Holden and Tilahun [7] found that these groups were quite well organized and formalized as primary cooperatives. They found that the groups to a large extent organized themselves in accordance with Elinor Ostrom's Design Principles (DPs) [8,9]. These eight DPs include clearly defined borders (DP1), matching appropriation and provision rules (DP2), collective choice arrangements (DP3), monitoring system (DP4), graduated sanctions (DP5), conflict resolution mechanism (DP6), recognized rights to organize (DP7), and nested enterprises (DP8). Holden and Tilahun [7] focused on the first six of these DPs as there was no variation in DP7 and DP8 in their sample. A set of performance indicators were positively correlated with their degree of compliance with these Design Principles [7]. One of these performance indicators was the perceived level of within-group trust. This study is a follow-up study of a sub-sample of 246 groups that were surveyed in 2019 with individual group member interviews and experiments to measure generalized as well as ingroup trust and social preferences of group members. Our study contributes to the literature on collective action and the importance of social preferences, norms of reciprocity and trust for the performance of groups that represent recently formed social-ecological systems (SES) [10]. Trust was among the ten second-level variables identified by Ostrom [10] to be crucial for the ability of groups to self-organize and is expected to enhance group cooperation [9,11]. It is believed that trust has important implications for the initiation, commitment, and longevity or dissolution of close relationships [12] and for the resilience of natural resource management institutions [13].

Trust, norms of reciprocity and social preferences represent forms of social capital and they may explain as well as be the result of development [14,15]. Other-regarding preferences are recognized to be important for economic and social outcomes such as cooperation in the workplace [16]. Our study builds on second-generation collective action theories which acknowledge that a significant proportion of individuals have non-selfish preferences [17–20]. Social motivations and endogenous preferences play important roles in second-generation collective action theories [21–26]. Ostrom and Ahn [15] sees trust as a core link between various forms of social capital and collective action. Trusting other people is risky and trust is based upon beliefs about the trustworthiness of others [27]. Repeated interactions are needed to verify the beliefs and the outcome of such verifications can affect the beliefs and thereby the level of trust over time in small groups. Even selfish individuals find it beneficial to be trustworthy in such situations when their reputation matters for their future outcomes [15].

There exists no consensus on how best to define and measure generalized and particularized trust. We follow Fehr [20] and Coleman [28] and define and measure trust as the sending behavior of trustors in the standard trust game [29]. And we define and measure trustworthiness by the returning behavior of trustees in the trust game. By varying the players that the trust game is played with we obtain measures of generalized trust (for unknown persons within the same district and that are from the same ethnic group) and particularized (ingroup—members of the same business group who know each other well based on frequent interactions) trust and trustworthiness. Camerer and Fehr [30] defined "social preferences" as how people rank different allocations of material payoffs to themselves and others. Fehr & Schmidt [31] classified theories of other-regarding preferences into models of outcome-based or distributional (social) preferences, models of interdependent (or "type-based" preferences, and models with intention-based reciprocity that differ from reciprocity-based and type-based social preferences. While we identify social preference types, we do not construct utility functions that imply a specific choice between the different models of Fehr & Schmidt [31]. By use of a set of simple binary incentivized dictator games, we elicit generalized and particularized social

preferences building on Fehr et al. [32] and Bauer et al. [33]. Rothstein [34] emphasizes the importance of norms in creating and maintaining generalized trust. We have included survey questions on the norm to reciprocate in our study and assess how this norm varies in the ingroup and outgroup settings and is related to social preferences, trustworthiness, and trust.

We build on the recommendation by Manski [35] for the study of endogenous social effects to collect more and richer data by combining experimental data with observed behavioral and perception data. There is still a shortage of studies that combine these three types of data although the number of experiments has increased, including field experiments. Our study is utilizing a large sample compared to most studies of experimental trust and is unique in assessing how generalized trustworthiness and trust are related to social preferences and norms of reciprocity and the formation of ingroup trustworthiness and trust in youth business groups. The composition of social preference types within groups turns out to explain much of the between-group heterogeneity in trustworthiness and trust.

The overall objective of this study is to examine the level of trust within these recently formed youth business groups and how it relates to generalized individual trust and social preferences and group performance. We aim to answer the following research questions. How do social preferences and norms of the youth group members influence outgroup and ingroup trust and trustworthiness? And, how much within-group and between-group variation is there in social preferences, norms of reciprocity and generalized trust and trustworthiness and does this affect trust-building within groups? How do social preferences, trustworthiness, and trust among youth in these groups compare to that found in other studies? To what extent can the good performance by the youth business groups be due to such preferences, norms, and trust and are these very different from that of youth in other places? This matters for whether the youth business group organizational model is likely to be transferable to other places.

The specific objectives are to a) assess the variation in individual outgroup and ingroup trust and trustworthiness and how these are related to social and economic preferences, social norms of reciprocity and social relations in the groups; b) assess the variation in group-level trust and how it is related to group characteristics in terms of the distribution of social preferences and norms, outgroup trustworthiness, and social relations in the groups; and c) assess the extent to which social preferences and norms enhance or constrain ingroup trust-building and group performance. Our findings are likely to be of high relevance for the sustainability of the youth group model in the study areas and for its generalized relevance to other places.

## Materials and methods

### Context: Northern Ethiopia

Population growth will continue to be high in Sub-Saharan Africa (SSA) for several decades and combined with climate change there will be a formidable policy challenge to create sustainable livelihood opportunities for the growing population. Much of the population growth will take place in rural areas. Creating youth employment is therefore high on the agenda in many SSA countries, including Ethiopia. There is a need to increase the absorption capacity of rural areas to limit rural-urban migration as well as international migration, which is becoming increasingly unpopular in receiving countries.

Land-use intensification and rural transformation are keys to enhancing the absorption capacity of rural areas, protecting the natural resource base, and creating sustainable livelihoods. A lot has been done in this direction in our study areas in Tigray Region in northern Ethiopia, which are characterized by a semi-arid climate with a long dry season and erratic rainfall. Large investments have been made in soil and water conservation, tree planting, and protection of the natural vegetation. Local collective action has played a central role with

support from the outside to halt land degradation and facilitate rehabilitation of large areas. Tigray Region received the Future Policy Gold Award 2017 from the World Future Council and the United Nations Convention to Combat Desertification (UNCCD) for its youth-inclusive land restoration policy [36]. This policy has for many years included a community-level approach to watershed management where all able-bodied adult members had to contribute 20–60 days per year of free labor for investment in local public goods. This has been combined with food-for-work and cash-for-work with additional funding from the outside such as from the UN World Food Program, The World Bank, and other donors.

To tackle growing rural landlessness the youth business group initiative we study was initiated around 2011 by the regional government [7]. Holden and Tilahun [7] made a census of 740 such groups in five districts in Tigray in 2016. The groups can be categorized into two main types, temporary mineral groups (about 300 of the groups) and groups provided rehabilitated communal lands to establish a sustainable natural resource-based business. The mineral groups were given a temporary license to extract a mineral resource to accumulate capital. When a target capital level has been reached, they "graduate" and are expected to find another livelihood where they can invest the starting capital they have raised as members of the mineral group. This study focuses on the other category of groups that are allocated more permanent rights to rehabilitated communal land areas. This study is based on data from 2427 youth business group members from 246 youth groups in four districts in Tigray region of Ethiopia. The study was conducted in January-May 2019 and up to 12 randomly sampled group members per group took part in interviews among those that were available. Our sample size is large and representative of the population of youth business groups in the study region. Details on the demographic characteristics of the sample youth group members is presented in Table 1. The median age of the sample youth group members was 31 years and 69% of them were male and the median number of school years completed was 4.

Formally, these groups are established as primary cooperatives based on local cooperative law. To be eligible the youth must be registered as landless or near landless in their home community (*tabia*) and apply to join the program. Group members have typically self-selected themselves within a neighborhood to form groups. The groups self-organize by electing a board of five members (leader, vice-leader, secretary, accountant, and treasury), establish their

**Table 1. Demographic characteristics of sample youth group members by district.**

| Woreda/District | Variable | Mean | St. err. of mean | Median | Number of members | Number of youth groups |
|---|---|---|---|---|---|---|
| Raya Azebo | Age | 32 .88 | 0 .416 | 33 | 482 | 47 |
| | Sex (1 = male, 0 = female) | 0 .66 | 0 .022 | 1 | | |
| | Education | 2 .81 | 0 .171 | 0 | | |
| Degua Temben | Age | 31 .68 | 0 .414 | 30 | 573 | 53 |
| | Sex (1 = male, 0 = female) | 0 .72 | 0 .019 | 1 | | |
| | Education | 4 .64 | 0 .155 | 4 | | |
| Seharti Samire | Age | 36 .53 | 0 .597 | 34 | 385 | 40 |
| | Sex (1 = male, 0 = female) | 0 .76 | 0 .022 | 1 | | |
| | Education | 3 .88 | 0 .188 | 3 | | |
| Adwa | Age | 30 .71 | 0 .246 | 29 | 987 | 106 |
| | Sex (1 = male, 0 = female) | 0 .65 | 0 .015 | 1 | | |
| | Education | 5 .87 | 0 .123 | 6 | | |
| Total | Age | 32 .30 | 0 .192 | 31 | 2427 | 246 |
| | Sex (1 = male, 0 = female) | 0 .69 | 0 .009 | 1 | | |
| | Education | 4 .66 | 4 .000 | 4 | | |

own bylaw, and make a business plan that has to be accepted by the local authorities. Their bylaws include rules for organizing group activities such as group meetings and group work activities, sharing rules for responsibilities and incomes, and punishment rules for violations. Their accounts are also subject to auditing by the local authorities. Some support and monitoring are provided by local youth associations. Some groups have benefitted from donations and have obtained credit for investments.

Each formally registered group is provided a demarcated area, typically a rehabilitated communal land area, for their activity. They are required to manage this area in a sustainable way and to protect the indigenous species growing there. At the same time, enrichment planting with e.g. eucalypts is allowed, and so is the planting of other trees and bushes and harvesting of grass as fodder for animals. Apiculture, livestock (cattle fattening, sheep and goat fattening, dairy), irrigation (vegetables and fruits), and forestry are the dominant group production activities on the allocated land areas [7].

Holden and Tilahun [7] found that the youth groups to a large extent complied with Ostrom's Design Principles and that their degree of compliance with these was positively correlated with group trust, group size stability, Youth Association ranking and group income per member. They assessed trust with a 5-level Likert scale ranking by the group leader.

## Theoretical framework and conceptual model

Ostrom [10] identified norms and social capital (moral and ethical standards regarding how to behave in groups including norms of reciprocity and trust) as one of ten crucial second-level (set of) variables that can reduce the transaction costs in reaching agreements and lower costs of monitoring [37–39].

Trust can be an important indicator of group performance and be associated with the characteristics of group members, their preferences, norms, and expectations that are crucial for solving collective action problems and making groups work better. The relations between individual social and economic preferences, norms, expectations, and behavior in the form of trust and trustworthiness are complex in closely-knit groups. We build on second-generation theories of collective action and take heterogeneous preferences seriously [15]. We, therefore, take social preferences as independent and non-reducible reasons why some individuals are more trustworthy than others and have stronger norms to reciprocate. Our basic assumption is that such preferences and norms and the distribution of these in groups can be important explanations for the building of within-group trust, which is an important basis for collective action [15]. We present a simple conceptual model (Fig 1) to illustrate the core parts of these relations. We later use group member data and group level data to empirically estimate these relations. We split individual group member preferences in social and economic preferences. Social preferences are outcome-based or distributional- and/or reciprocity-based preferences [32] while economic preferences include selfish preferences (lack of social preferences) and risk preferences. Risk preferences are relevant for trust as trusting people is risky.

We distinguish outgroup and ingroup trust and trustworthiness. For our purpose we define an "ingroup" as a real-world youth business group where all members know each other well and run a joint part-time business. Such groups may potentially develop high levels of (particularized) trust based on frequent interactions. By ingroup trust, we mean the level of trust that group members feel towards other (anonymous) members of their own group, and likewise for trustworthiness.

As a benchmark (generalized trust) within the specific society, we use the level of trust among the same group members towards unknown persons from the same district and ethnic group. We capture this by specifying the games as played with an anonymous member of

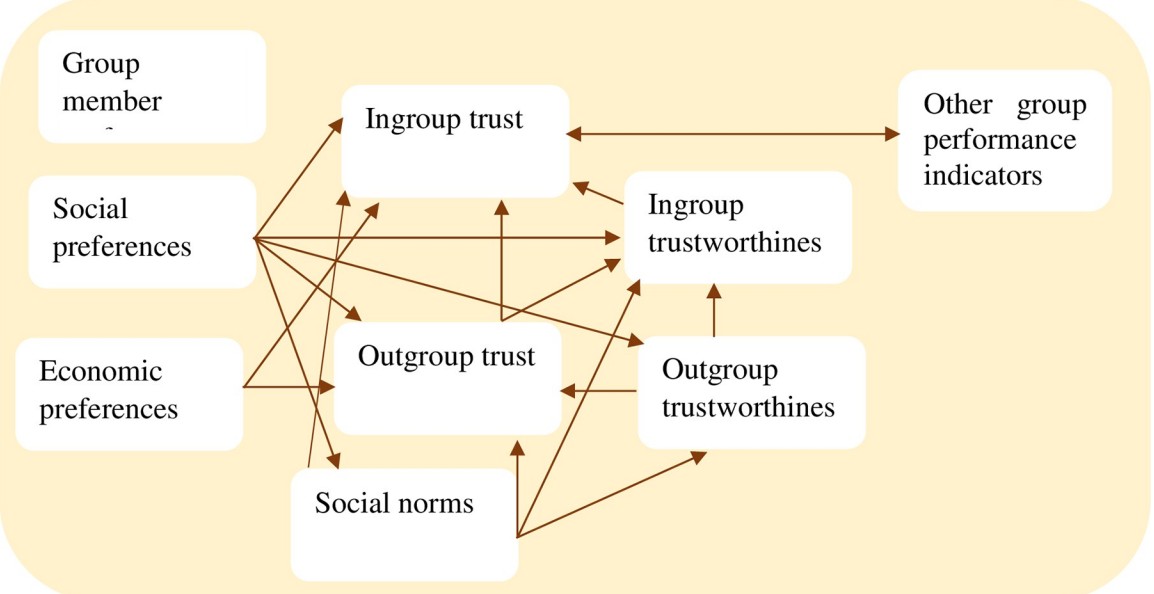

**Fig 1. Conceptual model.**

another unknown youth business group within the same district ("outgroup"). As youth business group members are resource-poor, we frame the outgroup as similarly resource-poor to avoid bias due to expected wealth differences. Generalized trust depends on social preferences, cultural norms, social stability, and many other factors that we do not aim to investigate here. We take social preferences as given individual characteristics. Carlsson et al. [40] found social preferences to be quite stable over longer periods of time. We allow social preferences to change with social distance and therefore to differ in the outgroup and ingroup contexts. We do the same for norms of reciprocity. We use three-level categorical variables to capture variation in norms of reciprocity in the generalized (outgroup) and particularized (ingroup) settings.

We use incentivized trust games [29] to get measures of ingroup and outgroup trust and trustworthiness. We also use incentivized experiments to reveal indicators of social and economic preferences of group members, building on Fehr et al. [32], Bauer et al. [33] and Gneezy and Potters [41]. We have identified members that are altruistic, egalitarian, spiteful or selfish in experiments with other unknown outgroup members. We assess how such social preferences may affect or be correlated with a norm to reciprocate, and thereby also affect individual trustworthiness as a basis for trust, both generalized (outgroup) trust, and particularized (ingroup) trust. Ostrom and Ahn [15] propose that dense social networks also enhance reciprocity norms through the transmission of information across individuals about who is trustworthy and who is not. We assess the extent of and difference in such norms of reciprocity in the outgroup (generalized) and ingroup contexts.

Repeated interactions within closely-knit groups hold the potential to build trustworthiness and trust within a short period of time but this depends on the ability of groups to function well. We use indicators for the social relations in the groups as additional indicators of group performance. Finally, we assess the correlation between ingroup trust and these other indicators of group performance. We expect high ingroup trust and trustworthiness to be positively related to the general social relations in the groups.

Trusting people is risky [27,42]. Economic preferences in terms of risk tolerance may therefore also play a role in determining outgroup and ingroup trust. Trust may therefore also depend on expected trustworthiness to the extent that trust has more selfish economic motivations. We have used a separate investment game based on Gneezy and Potters [41] to get measures of individual risk tolerance. We have also asked respondents about their expectations about the returns to their trust investments in the trust game. Together, risk tolerance and expected returns, may contribute to the explanation of the levels of outgroup and ingroup trust and the extent to which ingroup trust is higher than outgroup trust. We return to the more detailed model specifications and estimation strategy after we have outlined the experimental methods and data in more detail.

## Experimental methods and descriptive statistics

In this section, we outline the standardized experimental methods we applied to get measures of social preferences, trust, and trustworthiness. The detailed experimental protocols are presented in S1 Appendix. We also present descriptive statistics for the experimental outcomes.

**Ingroup versus outgroup trust games.** A binary stepwise version of the trust game [29] was used with a within-subject design where the group members in each case were offered 30 ETB that they could retain themselves or invest in another unknown person (see S1 Appendix for details of the game protocol). The respondents were asked how much they would be willing to invest when the other person; a) is an unknown person within their own group; b) is an unknown person in another youth group in the same district. The researchers triple the amount invested before it is given to the other person (trustee), who is free to return any amount to the trustor. The strategy method was used to obtain pre-committed amounts to be returned given varying amounts received as trustees. All sampled members played the roles as trustor as well as trustee. One of the games with the ingroup or the outgroup member was randomly drawn to become real.

Fig 2A shows the distribution of amounts invested in the trust game towards anonymous outgroup and ingroup members. Fig 2B shows the distribution of the individual ingroup net trust gain which is the ingroup minus the outgroup trust level for each group member. Trust is measured as the invested (trusted) share of the endowment provided. We see that very few respondents invested less in an anonymous ingroup member than in an unknown outgroup member. Most respondents invested substantially more in an ingroup member than in an outgroup member. We see this difference in trust as the gain in trust from group members knowing each other based on their frequent interactions over time and may be seen as a form of social capital that the group has achieved and that may be important for its performance of group activities. Summary statistics for key variables are presented in Table 2.

Fig 2A shows that there was a large difference in the ingroup versus outgroup trust. About 25% invested nothing in an outgroup member while less than 5% invested nothing in an anonymous ingroup member. The median amount invested in the ingroup trust game was the double of that invested in the outgroup trust game. The trustworthiness of outgroup trustees was limited; the majority returned a smaller amount than that sent by the trustors who sent some money. The median respondent only felt somewhat obliged to return an amount as large as that sent by an anonymous outgroup trustor.

**Social preferences and the norm of reciprocity.** Social preferences may contribute to explain trust and cooperation within groups as well as the behavior towards anonymous outgroup members. Building on the simple social preference games of Fehr et al. [20]; Fehr et al. [32]; Chowdhury et al. [43]; Bauer et.al. [33], we applied the extended version proposed by Bauer et al. [33] and classified respondents as altruistic, egalitarian, spiteful and selfish towards

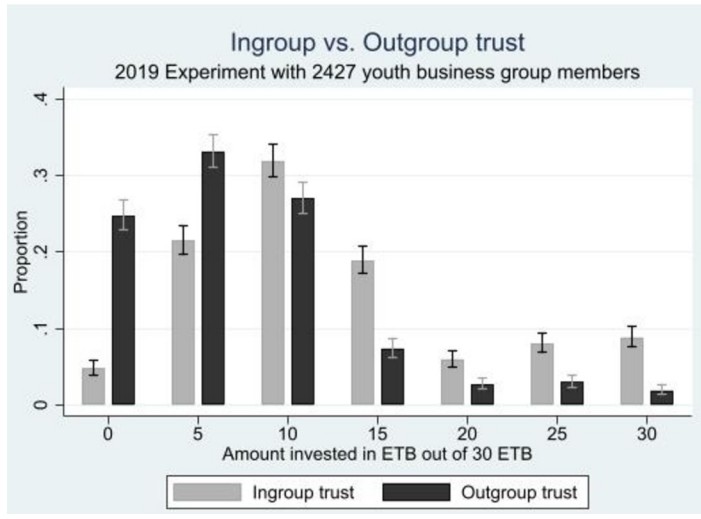
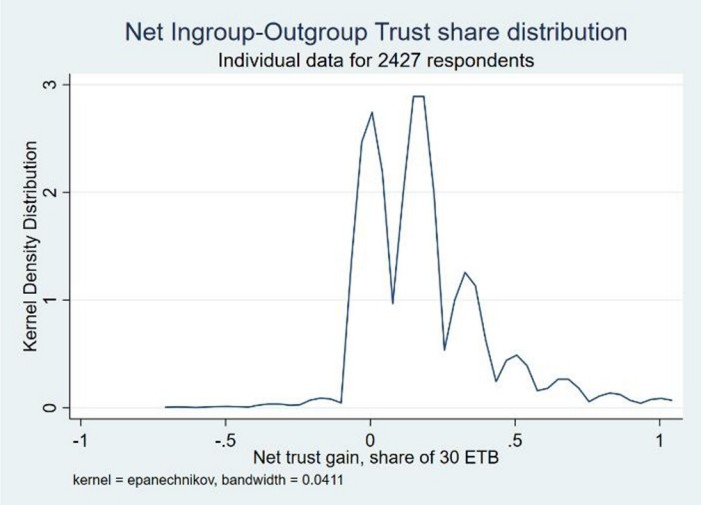

**Fig 2.** a. Ingroup versus outgroup trust investment. b. Net ingroup trust gain (share).

outgroup and ingroup members with the remaining being lumped together as one category with weaker preferences in these directions. The details of the experimental protocol are presented in S1 Appendix.

The set of experiments consists of four binary dictator games that each are played with an outgroup and an ingroup framing. Afterward, one is randomly chosen to become real. The games are: a) Costless prosocial game: Choice between (20 ETB (Ethiopian Birr), 20ETB) and (20ETB, 0ETB); b) Costless envy game: (20ETB, 20ETB) versus (20ETB; 40ETB); c) Costly prosocial game: Choice between (20ETB, 20ETB) and (40ETB, 0ETB); and d) Costly envy game: (20ETB, 20ETB) versus (30ETB, 40ETB) distribution between oneself and the other (outgroup or ingroup) player. Fehr et al. [32] used games a)-c) and Bauer et al. [33] added game d) that we also included. The classification into social preference categories is shown in Table A of S2 Appendix.

The social preference games were played before the trust games, but all payouts took place at the very end. The order of the games was the same for all respondents for practical reasons, which is a limitation of the study, and therefore did not allow us to test for order effects. The games were so simple that we do not expect much learning effects through the sequence of games, but we cannot rule out reflection effects [44] but all respondents are getting the same treatment in this respect.

A norm for reciprocation may be important for the extent to which respondents return money in the trust game. This norm may be an important determinant of own trustworthiness but may also affect expected trustworthiness and thereby trust.

In relation to outgroup anonymous trustors we asked the following question: As a receiver (trustee) in the game, how obliged do you feel to return an amount at least as big as the amount sent by the sender (trustor)? They had to choose among the following three responses: 3 = Extremely obliged, 2 = Somewhat obliged, 1 = Not obliged at all. Table 2 presents the responses for ingroup and outgroup players.

Table 2 provides a more detailed breakdown and shows that close to 32% feel extremely obliged to return an amount at least as large as the amount sent by the trustor in the outgroup trust game while 24% do not feel obliged at all, demonstrating substantial variation in the

**Table 2. Ingroup and outgroup reciprocity norm distribution.**

| | Ingroup | | Outgroup | |
|---|---|---|---|---|
| Norm | Freq. | Percent | Freq. | Percent |
| Extremely obliged | 1,448 | 59.7 | 764 | 31.5 |
| Somewhat obliged | 793 | 32.7 | 1,085 | 44.7 |
| Not obliged at all | 186 | 7.7 | 578 | 23.8 |
| Total | 2,427 | 100 | 2,427 | 100 |

*Source*: Youthbus baseline survey data 2019.

perception of this norm. In the ingroup context, 60% feel extremely obliged to return an amount at least as large as the amount sent by the trustor, demonstrating the strong group effect on the norm to reciprocate. It is only 8% that do not feel obliged at all to reciprocate in the ingroup context.

Table 3 presents more descriptive statistics and shows that about 10% of the respondents behave altruistically towards anonymous outgroup members compared to 25% for ingroup members, 17% (outgroup) and 18% (ingroup) behave in an egalitarian way (prioritize equal sharing), 33% (outgroup) and 28% (ingroup) behave selfishly and 17% (outgroup) and 3% (ingroup) behave in a spiteful way in the game. The remaining respondents, 24% (outgroup) and 26% (ingroup) express weaker preferences in these directions in the games. In the following econometric models, the latter group with weak social preferences is used as the reference base.

Bauer et al. [33] found in a sample of 4–12 years old children in the Czech Republic that 16% were altruistic, 9% inequality averse, 6% spiteful and 40% selfish. They found that spitefulness was associated with low education and poverty of parents. Fehr et al. [32] assessed these social preferences in 8–17 years old children in Tyrol, Austria. They found that spitefulness declines in frequency with age but was still more common than strong altruism and strong egalitarianism in 16/17-year-olds in ingroups as well as in outgroups of adolescents.

**Table 3. Summary statistics for key variables.**

| | Mean | Median | St. err. | Std. dev. |
|---|---|---|---|---|
| Ingroup trust, share invested | 0.413 | 0.333 | 0.005 | 0.265 |
| Outgroup trust, share invested | 0.227 | 0.167 | 0.004 | 0.216 |
| Net ingroup trust gain | 0.186 | 0.167 | 0.004 | 0.217 |
| Outgroup trustworthiness, share returned if receiving 30 ETB | 0.225 | 0.167 | 0.005 | 0.227 |
| Outgroup norm to reciprocate | 2.077 | 2 | 0.015 | 0.740 |
| Outgroup altruist dummy | 0.102 | | 0.006 | 0.302 |
| Outgroup egalitarian dummy | 0.167 | | 0.008 | 0.373 |
| Outgroup spiteful dummy | 0.167 | | 0.008 | 0.377 |
| Outgroup selfish dummy | 0.326 | | 0.010 | 0.469 |
| Ingroup trustworthiness, share returned if receiving 30 ETB | 0.315 | 0.333 | 0.005 | 0.225 |
| Ingroup norm to reciprocate | 2.520 | 3 | 0.013 | 0.635 |
| Ingroup altruist dummy | 0.252 | | 0.009 | 0.434 |
| Ingroup egalitarian dummy | 0.183 | | 0.008 | 0.387 |
| Ingroup spiteful dummy | 0.032 | | 0.004 | 0.175 |
| Ingroup selfish dummy | 0.277 | | 0.009 | 0.448 |

*Source*: 2019 Youthbus Baseline survey and experimental data for 2427 group members of 246 youth business groups.

The lower levels of education and more serious poverty in our sample than that of Bauer et al. [33] and Fehr et al. [32] have not made our sample relatively worse with respect to the distribution of these other-regarding preferences. We have about 3% of the members that were spiteful in the ingroup context and about 17% that were spiteful in the outgroup context.

*The distribution of social preferences across groups*. We have so far looked only at the overall distribution of preference and norm types in the outgroup and ingroup contexts. In addition, what is important in our study is to look at the variation in these distributions across groups. Fig 3A–3D present the variation in ingroup and outgroup social preference distributions. Fig 3A shows the distribution of altruists in the outgroup and ingroup contexts across groups. We see substantial variation across groups and particularly so in the ingroup context. This indicates that group members are more likely to behave altruistically towards ingroup members and more likely to be spiteful towards outgroup members. More altruistic preferences may also become "epidemic" within groups due to conditional reciprocation of altruism. Fig 3B shows that egalitarian preferences are more common in the outgroup context than altruistic preferences, but they are less likely to change when moving from the outgroup to the ingroup

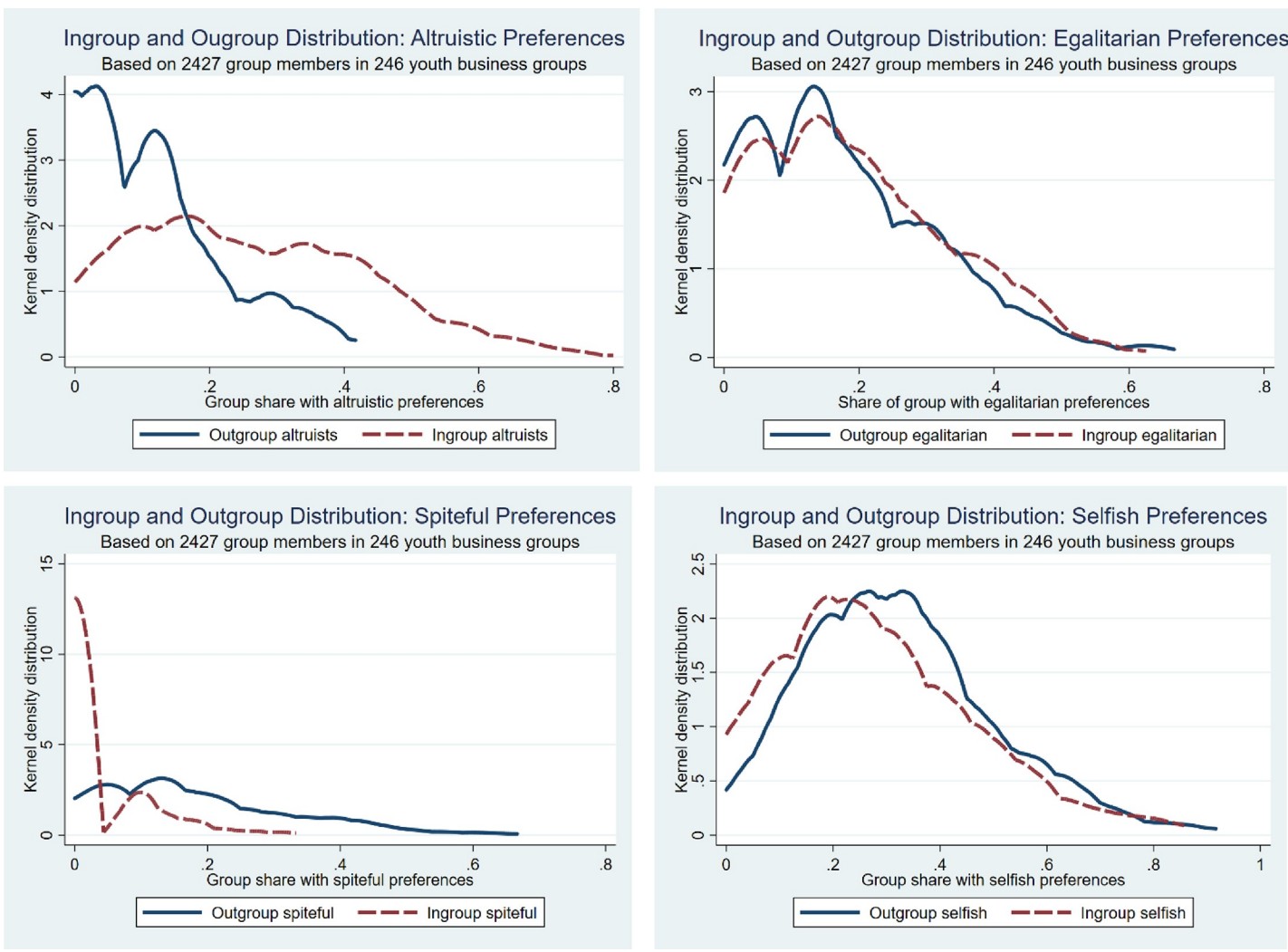

**Fig 3.** a. Ingroup and outgroup distribution of altruistic preferences. b. Ingroup and outgroup distribution of egalitarian preferences. c. Ingroup and outgroup distribution of spiteful preferences. d. Ingroup and outgroup distribution of selfish preferences.

context. Fig 3C shows that spiteful preferences are rare in both ingroup and outgroup contexts but there are a few groups with more spiteful members, particularly in the outgroup context. Fig 3D shows that selfish preferences are most common but the share with selfish preferences tends to be reduced in the ingroup context.

*Group level variation in trust and trustworthiness.* We are particularly interested in the across-group variation in trust and trustworthiness and how it relates to other group characteristics and their performance. We assess this by using group average responses from group members.

Fig 4A shows a substantial difference between ingroup and outgroup trust but also that there is a large variation in both these across groups and even that average ingroup trust in some groups is lower than outgroup trust in some groups. The group average net ingroup trust gain (the difference between average ingroup and outgroup trust) is also varying substantially but is positive for all groups. Fig 4B inspects the correlation between the ingroup and outgroup average trust measures, showing that they are quite strongly positively correlated. It indicates that when measuring ingroup trust we should take the outgroup trust into account. The net ingroup trust gain may be a better measure of the "trust effect" (social capital created) in the group than ingroup trust per se.

*Other group performance indicators.* Tables 4 and 5 give an overview of the additional group performance indicators that we have used. These include members' satisfaction with group leaders, group performance, and social relations in groups, and whether groups are perceived to be polarized/fractioned into sub-groups. 7.5% of the group members answered that their group was polarized and fractioned in sub-groups. Fig 5A–5E show the variation in these group performance indicators across the 246 groups.

## Model specifications and hypotheses

We consider outgroup trust as a measure of generalized trust among young adults that live under similar conditions in the same district. We use this as a benchmark (control) to assess ingroup trust that may depend on group performance and the social relations within groups on top of the factors that affect generalized trust and trustworthiness. We regard ingroup trust

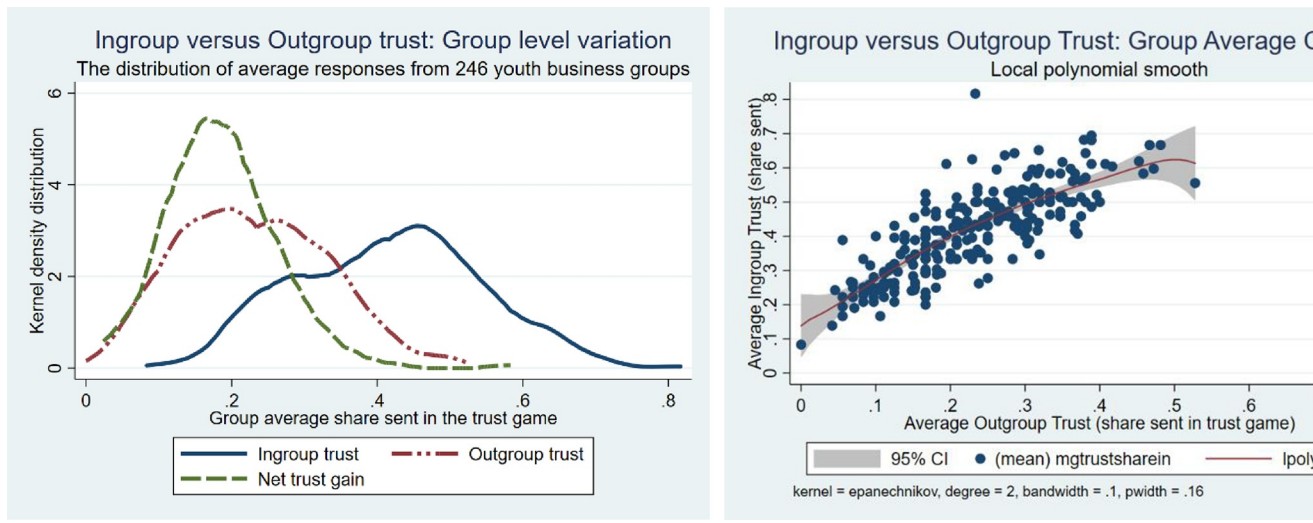

**Fig 4.** a. Average ingroup and outgroup trust and net trust gain. b. Average ingroup and outgroup trust correlation.

**Table 4. Group leader satisfaction and group performance since the beginning.**

| Satisfaction with group leader | | | Group performance since beginning | | |
|---|---|---|---|---|---|
| | **Freq.** | **Percent** | | **Freq.** | **Percent** |
| Very satisfied = 5 | 1,115 | 45.9 | Much improved = 5 | 164 | 6.8 |
| Quite satisfied = 4 | 689 | 28.4 | Improved = 4 | 1,458 | 60.1 |
| Acceptable performance = 3 | 311 | 12.8 | Stable = 3 | 551 | 22.7 |
| Not so satisfied = 2 | 61 | 2.5 | Declined = 2 | 200 | 8.2 |
| Very unsatisfied = 1 | 31 | 1.3 | Much declined = 1 | 54 | 2.2 |
| Leader | 220 | 9.1 | | | |
| Total | 2,427 | 100 | Total | 2,427 | 100 |

*Source*: Youthbus baseline survey data 2019. Leaders did not respond to the questions regarding their own performance.

as a group performance indicator [7]. Based on the conceptual model in Fig 1 we estimate the following models:

$$SNO_{gi} = oprob(SPO_{gi}, E_t) + e_{sno} \tag{1}$$

where *SNO* represents the outgroup norm to reciprocate in the trust game when playing it with unknown outgroup members. *SNO* is assumed to be partly a function of the outgroup social preferences (*SPO*) that we have measured, but also to have an independent individual component. We represent the social preferences by a dummy vector where each dummy variable represents members with altruistic, egalitarian, spiteful and selfish preferences with the remaining members with less strong preferences being the base category. The social norm is represented by the categorical variable with three levels; 3 = strong obligation to reciprocate, 2 = weak obligation to reciprocate, 1 = no obligation to reciprocate). We hypothesize that altruists and egalitarians have stronger norms for reciprocation than the base category (with weak social preferences) and that spiteful and selfish respondents have weaker norms for reciprocation than the base category. The $E_t$ variable is representing community (*tabia*) fixed effects as we assume community-level norms have such a locality nature. Next, we present a simple linear model for generalized individual (outgroup) trustworthiness and assume that it is influenced by social preferences and the norm for reciprocation.

$$TWO_{gi} = TWO_{gi}^0 + \alpha_{sp}SPO_{gi} + \alpha_{so}SNO_{gi} + E_c + e_{two} \tag{2}$$

where *TWO* represents individual outgroup trustworthiness which we hypothesize is

**Table 5. Social relations in groups and assessment of own performance.**

| Social relations in group ranking by members | | | How do you rate your own performance in the group from the beginning till today? | | |
|---|---|---|---|---|---|
| | **Freq.** | **Percent** | | **Freq.** | **Percent** |
| Very good = 5 | 569 | 23.4 | Much improved = 5 | 89 | 3.7 |
| Quite good = 4 | 1,370 | 56.5 | Improved = 4 | 1,536 | 63.3 |
| Ok = 3 | 450 | 18.5 | Stable = 3 | 640 | 26.4 |
| Not so good = 2 | 32 | 1.3 | Declined = 2 | 141 | 5.8 |
| Very bad = 1 | 6 | 0.3 | Declined = 1 | 21 | 0.9 |
| Total | 2,427 | 100.0 | Total | 2,427 | 100.0 |

*Source*: Youthbus baseline survey data 2019.

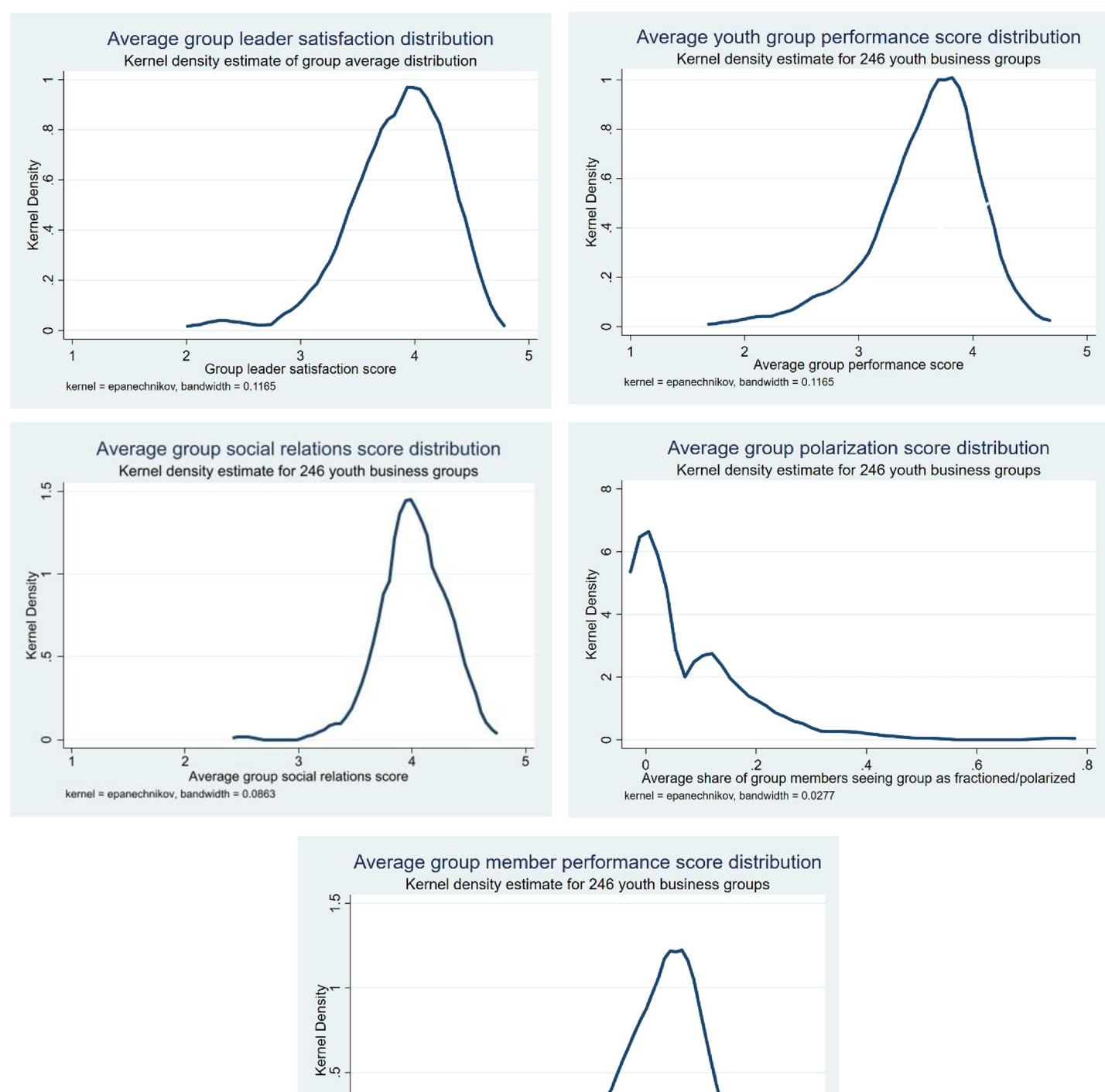

**Fig 5.** a. Average group leader satisfaction score distributions. b. Average youth group performance score distributions. c. Average group social relations score distributions. d. Average group polarization score distributions. e. Average group member performance score distributions.

enhanced by altruistic and egalitarian preferences and stronger social norms for reciprocation while spiteful and selfish individuals are hypothesized to demonstrate lower levels of generalized trustworthiness. We assess the endogeneity of the obligation to reciprocate by running this model without and with the social norm variable to assess how it affects the coefficients for the social preferences. This can reveal whether or to what extent the social preference variables operate through the norm or have a more direct effect. Additional controls are used for further robustness assessment (alternatively district or community fixed effects and experimental enumerator fixed effects). Next, we specify the model for generalized individual (outgroup) trust.

$$TO_{gi} = TO_{gi}^0 + \beta_{sp}SPO_{gi} + \beta_{so}SNO_{gi} + \beta_{two}TWO_{gi} + \beta_{exo}EXO_{gi} + \beta_r R_{gi} + e_{to} \tag{3}$$

Where $TO$ is outgroup trust, $EXO$ is the expected return from an unknown outgroup member and $R$ represents risk tolerance as additional economic preferences and expectations variables and we hypothesize that trust increases with expected return and risk tolerance. Expected return in the trust game is clearly endogenous and, ideally, we should estimate it separately. However, it is represented by a categorical variable that only partly is ordered. We have therefore chosen to include it directly in the main models presented. However, as a robustness check we endogenized expected returns in the outgroup and ingroup contexts with ordered probit models including the four first categories of the expected returns categorical variable. This implied a loss of observations in the estimation (n = 1703). The main results remained robust to this alternative specification in terms of parameter signs, although there were changes in the sizes and significance levels of some variables. The predicted expected return categorical variables were significant and with positive sign showing that expected returns matter for trust investment in both outgroup and ingroup settings.

Furthermore, we hypothesize that individuals with altruistic and egalitarian preferences trust more while selfish and spiteful individuals are less trusting, those with stronger norms for reciprocation trust more and so do the more trustworthy.

We now move to the ingroup models. We have the same logical sequence as for the outgroup models but in addition, assume that the outgroup models feed into ingroup responses. We also obtained data on ingroup social preferences and obligation to reciprocate. We model the ingroup social norm to reciprocate ($SNI$) on the ingroup social preferences ($SPI$) and assume that the outgroup social preferences influence through the predicted outgroup trust and trustworthiness variables. This helps to assess whether ingroup social preferences have a separate direct effect beyond what the outgroup social preferences have in the outgroup model structure. This depends on the degree to which ingroup social preferences differ from outgroup social preferences. The added value also depends on whether the ingroup social norm of reciprocation is different from that norm in the outgroup context. Our modeling approach allows us to test for such significant additional direct effects on ingroup trustworthiness ($TWI$) and trust ($TI$).

$$SNI_{gi} = oprob(SPI_{gi}) + e_{sni} \tag{4}$$

$$TWI_{gi} = TWI_{gi}^0 + \eta_{spi}SPI_{gi} + \eta_{sni}SNI_{gi} + \eta_{two}TWO_{gi} + \eta_{to}TO_{gi} + e_{twi} \tag{5}$$

$$TI_{gi} = TI_{gi}^0 + \mu_{spi}SPI_{gi} + \mu_{sno}SNI_{gi} + \mu_{two}TWO_{gi} + \mu_{to}TO_{gi} + \mu_{exi}EXI_{gi} + \mu_r R_{gi} + e_{ti} \tag{6}$$

We hypothesize that ingroup social preferences and social norms to reciprocate contribute to enhance ingroup trust. We also hypothesize that ingroup trustworthiness is enhanced by outgroup trustworthiness and trust. Furthermore, we hypothesize that ingroup trust-building

goes through the same channels as outgroup trust and is further strengthened through the formation of stronger ingroup social preferences and norms of reciprocity that also build ingroup trustworthiness. Finally, we hypothesize that ingroup trust also has an economic dimension as trusting people is risky and therefore more risk tolerant people invest more in the trust game and so do those with more optimistic expectations (*EXI*) about the return from their investment.

We acknowledge that multiple endogenous variables represent a formidable estimation challenge. However, we think that the six-equation recursive system goes far in capturing indirect endogenous effects. We do not claim that we have obtained fully unbiased and consistent estimates of the parameters. The structural equation model results can be assessed in relation to theory and be compared with the results from the naïve models with stepwise introduction of additional controls for consistency. The advantage of those models is that they reveal more about the explained within-group and between-group variation as additional controls are added (see results of models with group random effects in Tables C, E, and H of S2 Appendix). The functional form assumptions, as well as possible interactions and omitted latent variables, are likely to play a role.

## Estimation issues, data, and estimation strategy

Our data are such that we have two-stage sampling where groups were sampled first and then group members were sampled in the second stage. As groups are small, we must consider that data from group members are not independent and standard errors should be corrected for clustering at group level when analyzing individual-level data.

To a large extent, there was also self-selection of members into groups, and this could contribute to stronger ingroup social relations than outgroup social relations. We are to a limited degree able to separate this selection effect from the ingroup social relation formation effects after group formation. Many of the group members knew each other before they formed the group and they typically came from the same neighborhood (*got*) within the larger village (*kushet*) and municipality (*tabia*). The other selection criteria relate to the eligibility for joining a youth business group, which is related to being landless or very land-poor and being a resident of the *tabia* as well as aiming to establish a livelihood in the community and thereby demanding to join such a group. After joining, there is attrition that varies across groups as an additional selection mechanism, which could be influenced by many individual, group, community, and exogenous factors. We lack detailed data on dropped out members and cannot, therefore, assess the effect of this attrition.

As a potential indication of endogeneity of social preferences, we regressed the ingroup and outgroup preference variables on observable individual and parent characteristics while controlling for group fixed effects, see Table B in S2 Appendix. These characteristics explained very little of the variation in the data and few of the variables were significant even with such a big sample. This may indicate that most of the variation in the social preference variables can be regarded as exogenous. We therefore proceed based on that assumption.

Our empirical strategy is to assess how generalized trust and trustworthiness are related to basic social preferences and norms of reciprocity based on experimental measures of these where the youth group members played the games with unknown youth in other groups in their district. We assume that individual ingroup trust and trustworthiness also depend on these individual outgroup characteristics, complicating the analysis of ingroup trust and trustworthiness. We first do simple correlation analysis for the key variables and stepwise add variables as we move from one dependent variable to the next in the (recursive) conceptual model. We assess whether and to what extent adding variables increases the part of the variance that

can be "explained" and how the within-group and between-group variance is affected by the RHS variables in each model. We also assess the stability and statistical significance of the coefficients to get a first impression of their direct and potential indirect effects through added endogenous variables.

To deal with endogeneity, we run structural equation models in a recursive system based on the conceptual model. The identification strategy is as follows. We assume social norms are influenced at the community level and therefore use community (*tabia*) fixed effects in the ordered probit model for the social norm to reciprocate which has three levels (strong, weak and no obligation to reciprocate). The next level is outgroup trustworthiness, which was elicited with the strategy method by our experimental enumerators. This may have resulted in some enumerator bias in the data and we use enumerator dummies as instruments for identification. We had 12 enumerators to interview one group member each in each youth group. This was done both to ensure no communication among group members during experiments and interviews and to avoid spurious correlations within groups due to enumerator bias. For outgroup trust, we included additional economic preferences in the form of risk tolerance and outgroup expected returns in the game (a categorical variable). Trusting people is risky and more risk tolerant people are therefore expected to invest more, *ceteris paribus*, but more optimistic subjects with higher expected returns would also invest more.

For ingroup trustworthiness, we assume it is a function of outgroup trust and trustworthiness and use predicted values of these. In addition, we assume that ingroup social preferences and the ingroup norm of reciprocity affect ingroup trustworthiness. Ingroup norm of reciprocity is modeled on the ingroup social preferences with an ordered probit model, like the case of outgroup social norm was modeled on the outgroup social preferences. Ingroup trustworthiness is modeled on ingroup social preferences and the predicted ingroup norm of reciprocity. Finally, ingroup trust is modeled on the predicted ingroup trustworthiness, predicted ingroup social norm, predicted outgroup trustworthiness, predicted outgroup trust, ingroup social preferences, risk tolerance, and ingroup expected returns in the trust game.

To get further insights about within-group and between group variation, we ran single-equation models with random group effects that did not control for endogeneity or error correlations. These model results are included in S2 Appendix. We found substantial heterogeneity across groups making it interesting to study this further. We constructed the average group-level variables to dig deeper into the assessment of group effects.

The main advantage of this is that we can assess the group composition effects for social preferences. We, therefore, run models where the shares of altruistic, egalitarian, spiteful and selfish group members are included as additional variables that may influence individual norms, trustworthiness, and trust in the outgroup and ingroup contexts.

We included the shares of each social preference type in an alternative structural equation model. This implies a re-specification of model Eq 1 as follows:

$$SNO_{gi} = oprob(SPO_{gi}, \overline{SPO}_g, E_t) + e'_{sno} \qquad (1A)$$

and, likewise for Eqs 2–6. We hypothesize that these shares have a separate group effect on the dependent variables beyond the individual direct effects on their own norm, trustworthiness, and trust. We can then assess whether the outgroup and ingroup variation in norm, trustworthiness, and trust only are affected by the individual level social preferences or whether there is an additional effect of the frequency of or distribution of these norms in their group. We hypothesize that there are such group effects in the ingroup context but not in the outgroup context. More specifically, in groups with more altruists (egalitarians), we hypothesize that this has an additional positive effect on the ingroup norm to reciprocate, trustworthiness and trust.

Likewise, we hypothesize that ingroups with a larger share of spiteful and selfish ingroup members demonstrate a significantly lower level of the average ingroup norm to reciprocate, trustworthiness and trust. We have provided the datasets and a do file in STATA file formats as supporting information (see S1 and S2 Datasets, and S1 File). We acknowledge that there are alternative ways to estimate the system of equations and encourage others to inspect and scrutinize the robustness of our findings.

## Results

We started by assessing simple single equation models with latent group effects to assess the extent of within and between-group variation that can be explained by the included variables. These models are included in S2 Appendix for inspection by readers with interest in these. We also assessed the effect of including additional controls on the explained variation and on key parameters. A basic finding was that there was large across-group variation in social preferences, the norm of reciprocity, trustworthiness, and trust both in the outgroup and the ingroup contexts. The results were generally consistent with theory and with the structural equation models that we present below. We have, therefore, to save space, chosen to focus on these.

### Structural equation models: Combining outgroup and ingroup models

Table 6 presents the results of the base 6-equations structural equation model for generalized and particularized trustworthiness and trust, assuming that these are driven by social preferences and the norm to reciprocate. Trust is, in addition, assumed to be driven by risk tolerance and expectations. We highlight the following findings from Table 6. Social preferences in the form of altruistic and egalitarian preferences and the norm to reciprocate remain significant (with positive signs) throughout the outgroup models showing that these preferences are important for trustworthiness and trusting behavior. Spiteful and selfish individuals had significantly weaker norms of reciprocity and were less trustworthy and trusting. The norm to reciprocate is an important explanatory variable for generalized trustworthiness and trust and is stronger among individuals with altruistic and egalitarian preferences, and particularly so for the altruists. In contrast, the norm was weak for spiteful and selfish individuals and particularly so for spiteful individuals. Economic preferences (risk tolerance) and expected returns gave the theoretically expected and significant results, not very different from in the linear random effects models (see S2 Appendix). Overall, the results for generalized trust and trustworthiness remained robust and consistent across the linear random effects and the estimation using the structural equation model.

We inspected the direct versus indirect effects of outgroup social preferences on outgroup trustworthiness where the indirect effect goes through the norm to reciprocate by returning an amount at least as large as that sent by the trustor. This can be obtained from the two first equations in the system of equations where the norm to reciprocate equation is the first and is estimated with an ordered probit model. The results are summarized in Table 7.

Table 7 shows that altruistic, egalitarian and selfish preferences have significant direct and indirect effects on outgroup (generalized) trustworthiness while spiteful members have significant and strong indirect effects. The indirect effect of altruistic preferences on trustworthiness is substantially stronger than that of egalitarian preferences but both types of preferences pull in the same direction of enhancing outgroup trustworthiness.

The direct and indirect effects of selfish preferences are smaller in magnitude and of opposite sign (reducing outgroup trustworthiness) compared to that of altruists, while the total effects of spiteful preferences were strong and negative and mostly driven by a weak norm to reciprocate.

**Table 6. Structural equation model: Recursive system Outgroup -> Ingroup social preferences, norms, trustworthiness, and trust.**

| | (1) | (2) | (3) | (4) | (5) | (6) |
|---|---|---|---|---|---|---|
| | **Outgroup Norm to Reciprocate** | **Outgroup Trustworthiness** | **Outgroup Trust** | **Ingroup Norm to Reciprocate** | **Ingroup trustworthiness** | **Ingroup trust** |
| Altruist, dummy | 0.726*** | 0.046** | 0.134*** | 0.449*** | 0.014 | 0.034** |
| | (0.090) | (0.015) | (0.016) | (0.073) | (0.008) | (0.012) |
| Egalitarian, dummy | 0.215** | 0.044*** | 0.028* | 0.347*** | 0.010 | 0.009 |
| | (0.078) | (0.013) | (0.011) | (0.082) | (0.009) | (0.012) |
| Spiteful, dummy | -0.748*** | -0.021 | -0.022* | -1.216*** | -0.054** | -0.043* |
| | (0.086) | (0.012) | (0.009) | (0.150) | (0.019) | (0.018) |
| Selfish, dummy | -0.352*** | -0.038*** | -0.019* | -0.438*** | -0.015 | -0.026* |
| | (0.064) | (0.011) | (0.008) | (0.062) | (0.008) | (0.011) |
| Outgroup Norm to reciprocate, predicted | | 0.166*** | 0.033*** | | 0.061*** | 0.056*** |
| | | (0.006) | (0.006) | | (0.005) | (0.005) |
| Outgroup Trustworthiness, predicted | | | 0.223*** | | 0.665*** | 0.106*** |
| | | | (0.023) | | (0.022) | (0.028) |
| Outgroup Trust, predicted | | | | | 0.044** | 0.553*** |
| | | | | | (0.017) | (0.028) |
| Expected return: <1/3, base | | | | | | |
| One third, dummy | | | 0.012 | | | -0.029 |
| | | | (0.012) | | | (0.022) |
| Half, dummy | | | 0.056*** | | | 0.016 |
| | | | (0.013) | | | (0.022) |
| More than half, dummy | | | 0.051* | | | 0.041 |
| | | | (0.022) | | | (0.023) |
| Nothing as I sent nothing, dummy | | | -0.164*** | | | -0.158*** |
| | | | (0.011) | | | (0.023) |
| Nothing, although I sent some, dummy | | | 0.023 | | | 0.036 |
| | | | (0.016) | | | (0.028) |
| Risk tolerance | | | 0.034*** | | | 0.0301* |
| | | | (0.010) | | | (0.013) |
| Enumerator FE | No | Yes | No | No | No | No |
| Tabia FE | Yes | No | No | No | No | No |
| Constant | | 0.535*** | 0.227*** | | 0.246*** | 0.324*** |
| | | (0.018) | (0.018) | | (0.013) | (0.029) |
| N*** | 2427 | 2427 | 2427 | 2427 | 2427 | 2427 |
| var(trustworthiness) | | 0.031*** | | | 0.018*** | |
| | | (0.002) | | | (0.001) | |
| var(trust) | | | 0.0250*** | | | 0.038*** |
| | | | (0.001) | | | (0.002) |

*Note*: Six equations system model based on the Conceptual model in Fig 1. Estimated with GSEM in Stata. Standard errors are corrected for clustering at the youth group level. The social preference variables are for outgroups in models (1)-(3) and for ingroups in models (4)-(6). Outgroup and Ingroup social norm models are estimated as ordered probit models. Significance levels:

* $p<0.05$

** $p<0.01$

*** $p<0.001$.

**Table 7. Estimated direct, indirect, and total effects of social preferences and norms on outgroup trustworthiness.**

|  | Direct effect | Std. Err. | Indirect effect | Std. Err. | Total effect | Std. Err. |
|---|---|---|---|---|---|---|
| Outgroup norm to reciprocate | -0.166*** | 0.006 |  |  | -0.166*** | 0.006 |
| Outgroup Altruist, dummy | 0.046** | 0.015 | 0.121*** | 0.015 | 0.167*** | 0.023 |
| Outgroup Egalitarian, dummy | 0.044*** | 0.013 | 0.036** | 0.013 | 0.080*** | 0.019 |
| Outgroup Spiteful, dummy | -0.021 | 0.015 | -0.125*** | 0.015 | -0.145*** | 0.018 |
| Outgroup, Selfish, dummy | -0.038*** | 0.011 | -0.059*** | 0.011 | -0.097*** | 0.014 |

*Note*: Estimates based on the two-equation non-linear mediation model. The social nom model is estimated with ordered probit, the trustworthiness model is a linear model. The estimation is done in Stata 15.1 with the gsem command and indirect and total effects are estimated with the nlcom command.

We now turn to the ingroup models in Table 6. We first examine the results from the ingroup norm to reciprocate model. The signs for the social preference variables are consistent with that in the outgroup model and all are significant. Particularly, spiteful members disclosed a weak norm to reciprocate in the ingroup context. This demonstrates that the within-group norm to reciprocate is sensitive to the within-group variation in the distribution of these preferences. The fact that only the spiteful preference variable is significant (at 1% level) in the ingroup trustworthiness model (5) demonstrates that the indirect effect through the norm of reciprocation is most important (model (4)) and where all the preference variables were significant (at 0.1% levels).

We recall the substantial across-group variation in the distribution of social preferences in Fig 3A–3D. This may potentially explain a substantial share of the between-group variation in trustworthiness and trust and the effect to a large extent goes through the norm to reciprocate. The single equation models in S2 Appendix provide a clearer picture of this than the systems models. A weaker norm to reciprocate reduces ingroup trustworthiness less than it reduces outgroup trustworthiness but both these predicted effects are significant (at 0.1% levels). Likewise, the weak norm reduces ingroup trusting behavior.

We see that outgroup trustworthiness has strong predictive power and enhances ingroup trustworthiness. Individual generalized trustworthiness therefore also matters for ingroup trustworthiness. Predicted individual outgroup trust, in addition, has a positive and significant effect on individual ingroup trustworthiness.

Finally, we assess the ingroup trust model (6) which is the last equation in the system model. Table 6 shows that predicted outgroup trust and trustworthiness have significant positive effects on individual ingroup trusting behavior. The sizes of the coefficients for these predicted variables have been reversed compared to that in the ingroup trustworthiness model. Three of the ingroup social preference variables are significant. Spiteful and selfish individuals are less trusting and altruistic individuals more trusting than other members. Finally, we see that the expected returns and risk tolerance variables were less significant in the ingroup model.

So far, we have only assessed the individual preference and norm characteristics and their effects on individual trust and trustworthiness. We have not assessed how the variation in the within-group composition of these may indirectly affect the outgroup and ingroup individual trustworthiness and trust variables. To assess the extent of group effects from the variations in the compositions of the social preferences within groups we run the system of equations when also the within-group shares of altruists, egalitarian, spiteful and selfish members are included. We hypothesized that these are important for within-group trustworthiness and trust but not for generalized trustworthiness and trust. The model results are presented in Table 8.

The first model in Table 8, for the outgroup norm to reciprocate, we see that there are significant individual as well as group effects indicating that the outgroup norm to reciprocate is

**Table 8.  Structural equation model: With group mean proportions of social preference types and gender, age, and education.**

|  | (1) | (2) | (3) | (4) | (5) | (6) |
|---|---|---|---|---|---|---|
|  | Outgroup Norm to Reciprocate | Outgroup Trustworthiness | Outgroup Trust | Ingroup Norm to Reciprocate | Ingroup trustworthiness | Ingroup trust |
| Altruist, dummy | 0.618*** | 0.0407* | 0.128*** | 0.346*** | 0.012 | 0.025* |
|  | (0.095) | (0.016) | (0.016) | (0.076) | (0.008) | (0.013) |
| Egalitarian, dummy | 0.165* | 0.0404** | 0.030* | 0.206* | 0.008 | 0.009 |
|  | (0.083) | (0.014) | (0.012) | (0.082) | (0.009) | (0.013) |
| Spiteful, dummy | -0.611*** | -0.006 | -0.014 | -0.928*** | -0.0417* | -0.033 |
|  | (0.090) | (0.013) | (0.010) | (0.159) | (0.021) | (0.018) |
| Selfish, dummy | -0.252*** | -0.0229* | -0.005 | -0.329*** | -0.009 | -0.018 |
|  | (0.067) | (0.011) | (0.009) | (0.065) | (0.008) | (0.011) |
| Altruist share in group | 0.630* | 0.055 | 0.013 | 0.316 | 0.011 | 0.051 |
|  | (0.298) | (0.055) | (0.039) | (0.259) | (0.024) | (0.038) |
| Egalitarian share in group | 0.302 | 0.020 | -0.028 | 0.748** | 0.004 | -0.008 |
|  | (0.273) | (0.038) | (0.031) | (0.238) | (0.028) | (0.039) |
| Spiteful share in group | -1.005*** | -0.0844* | -0.070* | -2.184*** | -0.095 | -0.085 |
|  | (0.255) | (0.036) | (0.031) | (0.550) | (0.066) | (0.083) |
| Selfish share in group | -0.697** | -0.0824* | -0.087** | -0.623* | -0.035 | -0.043 |
|  | (0.222) | (0.034) | (0.027) | (0.243) | (0.027) | (0.036) |
| Outgroup Norm to reciprocate, predicted |  | 0.160*** | 0.028*** |  | 0.059*** | 0.053*** |
|  |  | (0.006) | (0.006) |  | (0.005) | (0.005) |
| Outgroup Trustworthiness, predicted |  |  | 0.215*** |  | 0.658*** | 0.099*** |
|  |  |  | (0.023) |  | (0.022) | (0.028) |
| Outgroup Trust, predicted |  |  |  |  | 0.0417* | 0.539*** |
|  |  |  |  |  | (0.018) | (0.029) |
| Expected return: <1/3, base |  |  |  |  |  |  |
| One third, dummy |  |  | 0.014 |  |  | -0.031 |
|  |  |  | (0.011) |  |  | (0.022) |
| Half, dummy |  |  | 0.058*** |  |  | 0.014 |
|  |  |  | (0.013) |  |  | (0.021) |
| More than half, dummy |  |  | 0.050* |  |  | 0.038 |
|  |  |  | (0.021) |  |  | (0.022) |
| Nothing as I sent nothing, dummy |  |  | -0.163*** |  |  | -0.160*** |
|  |  |  | (0.011) |  |  | (0.023) |
| Nothing, although I sent |  |  | 0.019 |  |  | 0.032 |
| Some, dummy |  |  | (0.015) |  |  | (0.028) |
| Risk tolerance |  |  | 0.032*** |  |  | 0.030* |
|  |  |  | (0.010) |  |  | (0.013) |
| Male, dummy | 0.184*** | 0.014 | 0.030*** | 0.104 | 0.011 | 0.027** |
|  | (0.051) | (0.008) | (0.007) | (0.055) | (0.006) | (0.009) |
| Age, years | 0.004 | -0.001* | 0.001* | -0.001 | -0.001*** | 0.000 |
|  | (0.003) | (0.000) | (0.000) | (0.003) | (0.000) | (0.000) |
| Education, years | 0.013 | 0.000 | 0.003*** | -0.006 | -0.001 | 0.000 |
|  | (0.007) | (0.001) | (0.001) | (0.008) | (0.001) | (0.001) |
| Enumerator FE | No | Yes | No | No | No | No |
| Tabia FE | Yes | No | No | No | No | No |

(*Continued*)

**Table 8.** (Continued)

| | (1) | (2) | (3) | (4) | (5) | (6) |
|---|---|---|---|---|---|---|
| | **Outgroup Norm to Reciprocate** | **Outgroup Trustworthiness** | **Outgroup Trust** | **Ingroup Norm to Reciprocate** | **Ingroup trustworthiness** | **Ingroup trust** |
| Constant | | 0.552*** | 0.261*** | | 0.253*** | 0.329*** |
| | | (0.028) | (0.025) | | (0.020) | (0.035) |
| Cut 1 | -0.049 | | | 0.278 | | |
| | (0.172) | | | (0.156) | | |
| Cut 2 | 1.349*** | | | 1.614*** | | |
| | (0.176) | | | (0.164) | | |
| N | 2427 | 2427 | 2427 | 2427 | 2427 | 2427 |
| var(trustworthiness) | | 0.0308*** | | 0.018*** | | |
| | | (0.002) | | | (0.001) | |
| var(trust) | | | 0.0248*** | | | 0.038*** |
| | | | (0.001) | | | (0.002) |

*Note*: Six equations system model based on the Conceptual model in Fig 1. Estimated with GSEM in Stata. Standard errors are corrected for clustering at the youth group level. Social preference variables are in the outgroup setting for models (1)-(3) and for the ingroup setting for models (4)-(6). Outgroup and Ingroup social norm models for obligation to reciprocate are estimated as ordered probit models. Significance levels:

* $p<0.05$

** $p<0.01$

*** $p<0.001$.

less exogenous than we hypothesized. The norm to reciprocate towards unknown persons is significantly stronger in groups with larger shares of altruistic members and significantly weaker in groups with a higher share of spiteful and selfish members. Particularly, the presence of more spiteful members in the group appears to strongly undermine the group norm to reciprocate.

Comparing the results in Tables 5 and 7, we see that the group mean effect in Table 8 on outgroup trustworthiness was captured as a direct effect in Table 6. We learn two things from this. The first is that the mechanism for change in the social norm to reciprocate is a group effect that varies with group composition in social preferences. This is showing that the generalized norm to reciprocate also can be influenced in the short to medium run in such small groups. The second is that what appeared as a direct effect also is partly an indirect group effect. We see that groups with a higher share of spiteful and selfish members are on average significantly less trustworthy and less trusting in the outgroup context, *ceteris paribus* (these effects are significant at 5% level).

Next, we look at the group composition effects in the ingroup context where we hypothesized to see such effects (unlike in the outgroup context). For the ingroup norm formation, we again find significant group effects in the same direction as in the outgroup model. The proportion of egalitarians enhances the likelihood that group members express a stronger ingroup norm to reciprocate. The proportions of spiteful and selfish group members have significant negative effects on this norm, like in the outgroup context, and the effect is particularly strong for spiteful members. We should recall, however, that there were few group members that were spiteful in the ingroup context.

The ingroup models show that the social preference group effects primarily work through changing the group member norms and thereby indirectly affect ingroup trustworthiness and trust. Table 8 models for ingroup trustworthiness and trust confirm the findings from Table 6

**Table 9. Correlations between ingroup trust and five other group performance indicators.**

| | (1) | (2) | (3) | (4) | (5) |
|---|---|---|---|---|---|
| | **trustsharein** | **Trustsharein** | **trustsharein** | **Trustsharein** | **trustsharein** |
| Average leader satisfaction score | 0.0518** | | | | |
| | (0.019) | | | | |
| Individual leader satisfaction score | 0.006 | | | | |
| | (0.003) | | | | |
| Average group performance score | | 0.004 | | | |
| | | (0.019) | | | |
| Individual group performance score | | 0.011 | | | |
| | | (0.007) | | | |
| Average group social relations score | | | 0.052 | | |
| | | | (0.029) | | |
| Individual group social relations score | | | 0.022* | | |
| | | | (0.008) | | |
| Average group polarization score | | | | -0.200** | |
| | | | | (0.076) | |
| Individual group polarization dummy | | | | -0.028 | |
| | | | | (0.021) | |
| Average group member performance score | | | | | 0.019 |
| | | | | | (0.023) |
| Individual group member performance score | | | | | 0.0208* |
| | | | | | (0.009) |
| Constant | 0.292*** | 0.546*** | 0.339** | 0.524*** | 0.621*** |
| | (0.078) | (0.079) | (0.123) | (0.019) | (0.094) |
| N | 2427 | 2427 | 2427 | 2427 | 2427 |

*Note*: Linear models with random group effects and enumerator fixed effects (left out from the table). Standard errors corrected for clustering at the group level. Significance levels:

* $p<0.05$

** $p<0.01$

*** $p<0.001$.

that generalized (outgroup) trustworthiness and trust are important drivers of ingroup trust-worthiness and trust.

Finally, we added three individual characteristics as additional controls in Table 8. These were gender, age, and education (years completed). Table 8 shows that men had a stronger generalized norm to reciprocate and were more trusting in the outgroup as well as ingroup contexts. Members with more education were significantly more trusting in the outgroup context but not in the ingroup context. Age was negatively correlated with trustworthiness in the outgroup as well as ingroup contexts. We examined whether a longer vector of individual and parent characteristics were correlated with the social preference variables in Table B of S2 Appendix and found that these three variables were significantly associated with some of the social preference variables although the degree of correlation was very low (as seen by the R-squares in Table B of S2 Appendix).

Next, we assess correlations between ingroup trust, and five (other) group performance indicators based on the assessment by individual youth group members, see Tables 4 and 5 and Fig 5A–5E for an overview and Table 9 for the correlations. For each of the other group

performance indicators, we have included the individual assessment as well as group average assessments. Table 9 shows that the average group leader satisfaction score is positively correlated with ingroup trust (significant at 1% level). Individuals who rate social relations higher are significantly (at 5% level) more trusting. Ingroup trust is significantly (at 1% level) lower in groups which are identified as polarized by a larger share of their members. Finally, individuals who rate their own performance in the groups better are also significantly (at 5% level) more trusting. All the significant variables, therefore, point in the expected direction.

## Discussion

The scope of our study was to contribute within a sub-set of variables that Ostrom [10] identified as among the deeper level sub-set of variables of likely high importance for whether groups are able to establish and sustain collective action and that can prevent a 'tragedy of the commons' outcome [3]. We have shown how this sub-set of variables relate internally and that altruistic, egalitarian, spiteful and selfish preferences are important for the degree to which individuals posit norms to reciprocate and are trustworthy and trusting. We have also demonstrated that particularized trust and trustworthiness within groups build on generalized trustworthiness and trust and that these are sensitive to the composition of social preference types within the groups. Finally, we also showed that another of the deeper-level variables, leadership (satisfaction with group leader) was positively correlated with ingroup trust.

Our study has demonstrated the importance of social preferences and norms to reciprocate for trustworthiness and trust. We may regard group members with selfish preferences as those most closely resembling *Homo economicus* and a substantial share of the respondents fall in this category although a part of this group becomes less selfish in the ingroup setting than in the outgroup setting.

Our study revealed that other-regarding preferences play an important role in the formation of norms of reciprocity, trustworthiness, and trust. Many members (25%) behaved altruistically in the ingroup setting while only 10% did so in the outgroup setting. We also found that the group composition of social preferences mattered as trust and trustworthiness were enhanced more in groups with a higher share of altruists and that this group composition effect materialized through the formation of stronger norms for reciprocity. The presence of spiteful and selfish members had the opposite effect and undermined the group norm to reciprocate both in the outgroup and ingroup contexts. Thus, the generalized (outgroup) norms of reciprocity and trust were endogenous in line with the hypotheses in Fig 1.

Several authors have argued that certain social-environmental systems may require a stronger element of other-regarding preferences to get more stable equilibria with sustained collective action [45,46]. We find in our study that about 3% of the members had spiteful preferences in the ingroup context and 17% in the outgroup context. This may be compared to what Fehr et al. [32] and Bauer et al. [33] found in Austria and the Czech Republic for kids and adolescents. Bauer et al. [33] found that children from families with low education were more spiteful, more selfish, and less altruistic. If we were to extrapolate from their study, we should expect to find more spiteful and more selfish members in our study because the level of education is lower than that in the study by Bauer et al. [33]. However, they studied children only up to the age of 12 and showed that the share of spiteful members declined with age (5% were spiteful at the age of 10–12) while the share with altruistic preferences increased with age. In Table B of S2 Appendix we show that selfish preferences were strongly negatively related to education while spiteful preferences were, surprisingly, positively related to education.

We do not have the basis to claim that our sample contains more altruistic members than is likely to be found elsewhere. What we can say, however, is that the average level of generalized

trust in our sample is low, even in the African context based on the meta-study by Johnson and Mislin [47], which found the average levels of trust and trustworthiness in Africa to be significantly lower than in other parts of the world. The average share sent in the trust game in studies in Africa that were reviewed by Johnson and Mislin [47] was 0.46, compared to 0.41 in the ingroup context and 0.23 in the outgroup context in our study. The large social heterogeneity in Africa gives reasons to critically question the representativeness of the few existing (small sample) studies summarized by Johnson and Mislin [47]. The average share returned (trustworthiness) in the African studies covered by Johnson and Mislin [47] was 0.32 compared to 0.32 in the ingroup context and 0.23 in the outgroup context in our study. One reason we find low shares sent and returned may be that our sample is particularly poor. Johnson and Mislin [47] also indicated that when respondents are both trustors and trustees, when it is random whether games will be real, and when the strategy method is used, the shares sent and returned are likely to be lower. These may therefore also be reasons for lower rates sent and returned in our study. Anyway, we think we can rule out that collective action only works in our study groups because group members are particularly trustworthy and trusting. The institutional rules such as the group bylaws established from the beginning are likely to be of high importance [7].

The structural equation model results were assessed in relation to theory and can be compared with the results from the single equation linear models with stepwise introduction of additional controls (see S2 Appendix). We find that the results with both approaches are remarkably consistent in terms of signs and significance levels of key variables. As a robustness check, we assessed how the results were affected when we used net ingroup trust gain (= individual ingroup trust–individual outgroup trust) instead of ingroup trust. Almost all the results remained identical, with one exception. The net ingroup trust gain was negatively correlated with outgroup trust. Therefore, in groups with members with high levels of generalized trust, there may be less hope to further increase ingroup trust. This may be because the measure of trust is in the range 0–1.

We chose a cautious approach to assessing the relationship between ingroup trust and other group performance indicators. Based on the theory we expect positive correlations in terms of ingroup trust being positively correlated with other group performance indicators. We recognize that trust is endogenous like other performance indicators and these may be jointly determined by other observable and unobservable variables [20]. Fehr [20] stated that he has not seen any convincing studies of how trust affects other variables given the endogeneity issue and the difficulty of finding valid instruments for trust that would enable identifying its causal impacts. In our study, we do not aim to identify the causal effect of ingroup trust on group performance as we have not been able to think of any valid instruments.

Nevertheless, there are studies that have attempted to assess the impacts of trust on the group or team performance. De Jong et al. [48] state that trust in team (group) members has long remained a relatively neglected issue in research on trust in teams and has received less attention than trust in leadership [49–51]. In their meta-study of trust in teams, De Jong et al. [48] find that the effect of trust in team members is stronger than e.g. the effect of trust in team leadership. De Jong et al. [48] and Fulmer and Gelfand [51] recommend more research on trust in team members as a fruitful direction of future research. Our experimental approach to measuring trust in groups is essentially a contribution in this direction. We have used individual group members' measures of trust and used these to generate group-level measures of trust. We have run an additional set of models for the aggregate group-level variables (included in S3 Appendix). The results from these models are consistent with the findings from the individual-level models.

The fact that all the groups studied here have survived up to the time of our study while our study disclosed large variation in trust across groups, indicates that high trust is not a necessary condition for the short-term survival of these groups. Other studies have shown that freeriding is less likely to take place when there is communication among the parties [30,52,53] or where there is a likelihood that free-riders will be punished [30,54]. The group bylaws imposing compulsory frequent meetings and punishment rules for violations may be two of the key institutionalized rules that have contributed to group survival even for groups with low levels of trust [7]. Cook et al. [55] also argue that societies may function well without trust. Other institutional or organizational arrangements are then needed that serve as substitutes for trust. Further work is needed to study whether punishment can substitute for trust or can serve to enforce the norm of reciprocity [56].

## Conclusions

Our study of social preferences, the norm of reciprocity and trust in youth business groups in northern Ethiopia has demonstrated that substantial shares of poor and young rural youth exhibit other-regarding preferences and norms of reciprocity both in the generalized (outgroup) and particularized (ingroup) contexts. However, the average levels of ingroup and outgroup trust and trustworthiness revealed through the experimental trust games were low even in the African context and we found substantial heterogeneity in these characteristics across groups. Altruistic and egalitarian preferences were associated with stronger norms to reciprocate, higher outgroup and ingroup trustworthiness and trust while spiteful and selfish preferences had opposite effects. On average 10% of the members exhibited altruistic preferences in the outgroup context against 25% in the ingroup context, while the share with selfish preferences was 33% in the outgroup context and 28% in the ingroup context. 17% were spiteful in the outgroup context compared to 3% in the ingroup context. We found that the social preferences had both direct and indirect effects on trustworthiness and trust and that the norm to reciprocate was sensitive to the group composition of social preferences not only in the ingroup context but also in the outgroup context.

An overall important conclusion from this study is that the youth group model seems robust to substantial variation in social preferences, norms of reciprocity and trust within groups as all the groups included in this study have survived and members of most groups are satisfied with how the groups perform. Still, this does not mean that social preferences, norms, and trust do not matter. We found that ingroup trust was positively correlated with a number of group performance indicators. We may conclude that the apparent success and stability of this youth business group model is not due to the unique social preferences and particularly high levels of trustworthiness and trust in these youth groups. This may indicate that the model is transferable to other places in Africa with similar levels of trust and norms of reciprocity. What may be more important is the compliance with Ostrom's Design Principles as found by Holden and Tilahun [7] for these youth business groups. Still, social preferences and norms are important to enhance group performance and could in specific marginal situations be the factors that cause groups to collapse or survive under strong pressures. However, a more longitudinal study will be needed to learn more about this.

## Supporting information

**S1 Appendix. Experimental protocols.**
(DOCX)

**S2 Appendix. Social preference games and single equation models.**
(DOCX)

**S3 Appendix. Group level models: Social preferences, trust, and group performance.**
(DOCX)

**S1 Dataset. Dataset of social preferences of youth groups.**
(DTA)

**S2 Dataset. Dataset of social preferences of youth group members.**
(DTA)

**S1 File. Do-file for the econometric analyses of S1 and S2 Datasets.**
(DO)

## Acknowledgments

This research has been conducted as a collaboration between Norwegian University of Life Sciences (NMBU) and Mekelle University. The authors acknowledge good support from local government authorities, local Youth Associations, and Mekelle University, and committed efforts by our team of enumerators and field supervisors.

## Author Contributions

**Conceptualization:** Stein T. Holden.

**Data curation:** Stein T. Holden, Mesfin Tilahun.

**Formal analysis:** Stein T. Holden.

**Funding acquisition:** Stein T. Holden.

**Investigation:** Stein T. Holden, Mesfin Tilahun.

**Methodology:** Stein T. Holden.

**Project administration:** Stein T. Holden, Mesfin Tilahun.

**Resources:** Stein T. Holden, Mesfin Tilahun.

**Software:** Stein T. Holden.

**Supervision:** Stein T. Holden, Mesfin Tilahun.

**Validation:** Stein T. Holden, Mesfin Tilahun.

**Visualization:** Stein T. Holden, Mesfin Tilahun.

**Writing – original draft:** Stein T. Holden, Mesfin Tilahun.

**Writing – review & editing:** Stein T. Holden, Mesfin Tilahun.

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
