## [Decision Letter · Decision Letter 0]

1 Jul 2021

PONE-D-21-15257

Preferences, trust, and performance in youth business groups

PLOS ONE

Dear Dr. Gelaye,

Thank you for submitting your manuscript to PLOS ONE. After careful consideration, we feel that it has merit but does not fully meet PLOS ONE’s publication criteria as it currently stands. Therefore, we invite you to submit a revised version of the manuscript that addresses the points raised during the review process.

You will find two reports pasted to my message. Both reviewers liked the paper, the data and are quite positive regarding the paper but consider that the paper needs substantial re-writing. I agree with them that a more focused paper is always more desirable. I would strongly recommend to amend the paper following reviewers recommendations.

Note that I will send back the paper to the very same referees. Please submit your revised manuscript by Aug 13 2021 11:59PM. If you will need more time than this to complete your revisions, please reply to this message or contact the journal office at plosone@plos.org. Please include the following items when submitting your revised manuscript:

We look forward to receiving your revised manuscript.

Kind regards,

Pablo Brañas-Garza, PhD Economics

Academic Editor

PLOS ONE

Journal Requirements:

2. Thank you for including your ethics statement: “ The ethical standars were requirments when we submitted the proposal for The Research Council of Norway, and the fact that we got funding is because the proposal fulfills these ethical standards in human subject research. Moreover, data were analyzed anonymously. "

Please amend your current ethics statement to confirm that your named institutional review board or ethics committee specifically approved this study.

3. In order to improve reporting, in your methods section, please provide additional information about the participant recruitment method and the demographic details of your participants, such as a) the recruitment date range (month and year), b) a description of any inclusion/exclusion criteria that were applied to participant recruitment, c) a table of relevant demographic details, d) a statement as to whether your sample can be considered representative of a larger population, e) a description of how participants were recruited, and f) descriptions of where participants were recruited and where the research took place.

4. Could you please provide the online URL for the working paper? Thank you.

Reviewers' comments:

Reviewer's Responses to Questions

**Comments to the Author**

1. Is the manuscript technically sound, and do the data support the conclusions?

Reviewer #1: Partly

Reviewer #2: Partly

2. Has the statistical analysis been performed appropriately and rigorously? 

Reviewer #1: Yes

Reviewer #2: I Don't Know

3. Have the authors made all data underlying the findings in their manuscript fully available?

Reviewer #1: No

Reviewer #2: No

4. Is the manuscript presented in an intelligible fashion and written in standard English?

Reviewer #1: Yes

Reviewer #2: Yes

5. Review Comments to the Author

Reviewer #1: General evaluation of the paper:

The paper presents the results of an interesting experiment run in Ethiopia. The objective of the experiment is to elicit social preferences, trust, and trustworthiness of a very large sample (2427 individuals from 246 different cooperative business) to understand the characteristics of individuals and groups that can explain the success or failure or the cooperative group. The cooperative initiatives are part of a public program for fighting against poverty, giving opportunities to young people for running a job and generate income. The authors used a trust game with real incentives where everyone could be trustor and trustee, having the opportunity to share money with outgroup (other cooperative projects) and ingroup (same cooperative project) individuals. The usability of this paper is the possibility to run the same experimental model in other cooperative projects to predict success or failure of the group and prevent and predict future problems for the good functioning of the groups.

In my opinion, the article is clear and easy to read. Tables and figures are useful and informative. However, there are some issues with the article, which in my opinion means is not publishable in this moment.

Major comments:

Although there is a good list of references included in the article, I find some weaknesses in the presentation of literature that gives a good support to some aspects.

I should recommend presenting briefly the variables included in Elinor Ostroms’s Design Principles to better understand the election of trust and trustworthiness in the study. In addition, and what I consider more relevant, is that there are several assumptions in the model specifications and hypotheses that would be worth supporting by the literature as well as Ostrom’s theory. There are different and no convergent theories that explain outgroup and/or ingroup trust, preferences, and cooperation. So maybe it could be interesting to present some alternatives in the discussion.

Minor comments:

Line 293: there is an error in the number of the table referred. It is table number 1.

Line 302: repetition study-study.

Line 327: there is an error in the number of the figure. It is figure number 4b.

Line 681: is it not quite bold to consider "an African context"?

Reviewer #2: This manuscript touches off a very interesting topic and presents very powerful and rich data. The arguments and the conceptual model tested seem meaningful and thoughtful. However, there is huge room for improvement. On the one hand, I would suggest the authors to reduce as much as possible the main text and analyses in order to help the reader focus on the key elements. Currently, the paper seems unnecessarily long and complex to me. On the other hand, there are many typos, unorthodox expressions, and unclear parts, which make reading not to flow easily. I spot some of them below but my recommendation is to exert more effort and care in writing (which is not bad overall) and in structuring the flow of arguments. Below, I list more specific comments in (approx.) order of appearance in the text.

1. Abstract (and elsewhere): “Generalized” trust typically refers to unaffiliated individuals, not to outgroups (who would also be “particularized”). See, e.g., Brewer, M. B. (1999). The psychology of prejudice: Ingroup love or outgroup hate?. Journal of social issues, 55, 429-444. Please check throughout the manuscript and justify the use of language.

2. Abstract: please check “Our study has used incentivized”, better just “used”?

3. Line 39-40: “The pressures… is increasing” please check.

4. Line 83: I would suggest referring to “outcome-based social preferences” or “distributional (social) preferences” to distinguish from reciprocity-based and type-based social preferences. See, e.g., Fehr, E., & Schmidt, K. M. (2006). The economics of fairness, reciprocity and altruism–experimental evidence and new theories. Handbook of the economics of giving, altruism and reciprocity, 1, 615-691.

5. Figure 1 is a bit messy. I encourage the authors to think carefully how to improve visualization. It also adds economic preferences on top of social preferences but there was not previous mention to economic preferences. It is also unclear why individual preferences do not impact on group performance. In addition, it is unclear why economic preferences are not labeled as “risk preferences”.

6. Figure 2a: it is unclear what the error bars represent. Standard errors? Confidence intervals? Please clarify.

7. Figure 2b and elsewhere: “net trust gain” is an unorthodox label for the standard “ingroup favoritism in trust”, “ingroup bias in trust”, or “intergroup bias in trust”. The problem is that the label is misleading (why “gain”?). Also, please clarify why the kernel density plots use apparently different bandwidth specifications.

8. Line 263: “The order of the games was the same for all respondents for practical reasons”. Please list more clearly this as a limitation of the study.

9. Table B: although it is not impossible, it seems strange that age is positively related to both altruistic and spiteful preferences (ingroup) which are exactly the opposite preferences. Please mention about this result and why it is not strange.

10. Figure 3 (results on ingroup favoritism in social preferences): these results might be compared to those of Espín et al. (2019) where a similar analysis was performed. Espín, A. M., Correa, M., & Ruiz-Villaverde, A. (2019). Patience predicts cooperative synergy: The roles of ingroup bias and reciprocity. Journal of Behavioral and Experimental Economics, 83, 101465.

11. Line 307: “More altruistic preferences may also become “epidemic” within groups.” I do not understand what “epidemic” means in this context.

12. Line 310: please check “are rare in both ingroup context but”.

13. Line 326: “group average net trust gain (the difference between average outgroup and ingroup trust)” Shouldn’t be the opposite? i.e., difference between ingroup and outgroup.

14. Line 371: it is unclear what the “Z variable” refers to.

15. Line 528: please check “the results for generalized trust and trustworthiness the results”.

16. Line 565 and elsewhere: significance should not be qualified as “highly” (or any other qualification). Line 582: “less significant” is not a correct expression either. Please check throughout the manuscript.

17. Table 5 and elsewhere: “System of equations model” does not seem the most standard label for “structural equation model”.

18. Table 7: it seems that (at least in my pdf) the variable “egalitarian, dummy” is missing. Also, here and elsewhere, the “norm to reciprocate” variable seems defined in the wrong way (line 368). Please define it as a positive function of reciprocation norm, not as a negative function.

19. Line 665: “Thus, generalized norms of reciprocity and trust were less exogenous than we had hypothesized.” I do not understand this statement (Figure 1 shows a causal effect of social preferences on trust and social norms). Please clarify.

20. Lines 669-678: the aggregate % of each type might be meaningfully comparable to those of Corgnet et al. (2015) for example (at least an adult sample)? Corgnet, B., Espín, A. M., & Hernán-González, R. (2015). The cognitive basis of social behavior: cognitive reflection overrides antisocial but not always prosocial motives. Frontiers in Behavioral Neuroscience, 9, 287.

21. Line 698: “The net trust gain was negatively correlated with outgroup trust.” Of course, since “net trust gain” is defined as the difference between ingroup and outgroup trust. This seems misleading.

22. As a general comment, how can we be sure that the causal relationships are not different from those proposed in Figure 1? Is there any way to compare against alternative specifications?

6. PLOS authors have the option to publish the peer review history of their article (what does this mean?). If published, this will include your full peer review and any attached files.

Reviewer #1: No

Reviewer #2: No

---

## [Author Response · Author response to Decision Letter 0]

11 Aug 2021

I. Response to Editor’s Comments

Journal Requirements:

 1. Please ensure that your manuscript meets PLOS ONE's style requirements, including those 

for file naming. The PLOS ONE style templates can be found at

Response: We have strictly followed PLOS ONE's style requirements, including those for file naming in preparing the manuscript and supporting information.

2. Thank you for including your ethics statement: “The ethical standars were requirments when we submitted the proposal for The Research Council of Norway, and the fact that we got funding is because the proposal fulfills these ethical standards in human subject research. Moreover, data were analyzed anonymously.”

Please amend your current ethics statement to confirm that your named institutional review board or ethics committee specifically approved this study.

 Response: We have amended the ethical statement and stated that RCN has approved this study as follows. The ethical standards were requirements when we submitted the proposal for this study to The Research Council of Norway, and the fact that we got funding for this study is because the study fulfills these ethical standards in human subject research. Moreover, data were analyzed anonymously.

3. In order to improve reporting, in your methods section, please provide additional information about the participant recruitment method and the demographic details of your participants, such as a) the recruitment date range (month and year), b) a description of any inclusion/exclusion criteria that were applied to participant recruitment, c) a table of relevant demographic details, d) a statement as to whether your sample can be considered representative of a larger population, e) a description of how participants were recruited, and f) descriptions of where participants were recruited and where the research took place.

Response: We have provided additional information about the method we used in selecting the samples and a table on demographic characteristic of the sample of youth group members.

 4. Could you please provide the online URL for the working paper? Thank you.

Response: Here below are the links to the working paper. 

https://www.nmbu.no/download/file/fid/40164

https://ideas.repec.org/p/hhs/nlsclt/2019_008.html

Upon re-submitting your revised manuscript, please upload your study’s minimal underlying data set as either Supporting Information files or to a stable, public repository and include the relevant URLs, DOIs, or accession numbers within your revised cover letter. For a list of acceptable repositories, please see http://journals.plos.org/plosone/s/data-availability#loc-recommended-repositories. Any potentially identifying patient information must be fully anonymized. We will update your Data Availability statement to reflect the information you provide in your cover letter.

Response: We have organized the minimal underlying dataset and the econometric estimation methods in STATA files as supporting information in the revised version (See S1 Dataset of youth groups, S2 Dataset of youth group members, and S1 Do-file for S1 and S2 Datasets). 

II. Response to Reviewers’ Comments

Reviewer #1: General evaluation of the paper:

The paper presents the results of an interesting experiment run in Ethiopia. The objective of the experiment is to elicit social preferences, trust, and trustworthiness of a very large sample (2427 individuals from 246 different cooperative business) to understand the characteristics of individuals and groups that can explain the success or failure or the cooperative group. The cooperative initiatives are part of a public program for fighting against poverty, giving opportunities to young people for running a job and generate income. The authors used a trust game with real incentives where everyone could be trustor and trustee, having the opportunity to share money with outgroup (other cooperative projects) and ingroup (same cooperative project) individuals. The usability of this paper is the possibility to run the same experimental model in other cooperative projects to predict success or failure of the group and prevent and predict future problems for the good functioning of the groups.

In my opinion, the article is clear and easy to read. Tables and figures are useful and informative. However, there are some issues with the article, which in my opinion means is not publishable in this moment.

Response: Thank you for the positive feedback on the merit of the paper and we have done our best to address the comments and issues the reviewer has pointed out as described below.

Major comments:

Although there is a good list of references included in the article, I find some weaknesses in the presentation of literature that gives a good support to some aspects.

I should recommend presenting briefly the variables included in Elinor Ostrom’s Design Principles to better understand the election of trust and trustworthiness in the study. In addition, and what I consider more relevant, is that there are several assumptions in the model specifications and hypotheses that would be worth supporting by the literature as well as Ostrom’s theory. There are different and no convergent theories that explain outgroup and/or ingroup trust, preferences, and cooperation. So maybe it could be interesting to present some alternatives in the discussion.

Response: We have included in brief the variables of the Elinor Ostrom’s Design Principles. As we write, trust was one of the second-level variables that Ostrom identified as keys to successful collective action. We are not sure what exact alternative theories the reviewer has in mind. We received some suggestions from the other reviewer for additional references and have taken those into account. The second reviewer asked for simplification rather than elaboration and so did the editor, so we feel we are pulled in opposite directions here. There are certainly many points that could have been elaborated more with reference to different theories. However, it may add complexity rather than help simplify and shorten the paper. We would, however, be very happy to include any studies or theories that can help us build a stronger paper and analysis. 

Minor comments:

Line 293: there is an error in the number of the table referred. It is table number 1.

Response: Corrected and we put the paragraph as well next to Table 1 (Now table 2 in the revised version; see line 313 in the revised version). Note: we added one table on the demographic characteristics of the samples. Therefore, the table 1 in the earlier draft is now Table 2 in the revised version. 

Line 302: repetition study-study.

Response: Corrected (See line 344 in the revised version)

 Line 327: there is an error in the number of the figure. It is figure number 4b.

Response: Corrected (See line 370 in the revised version)

Line 681: is it not quite bold to consider "an African context"?

Response: The reviewer may have a good point here. Africa is very heterogenous and the few studies of trust and trustworthiness in Africa that were included in the meta-study by Johnson and Mislin may not be representative of average levels or the heterogeneity, given the relatively small samples and few locations where such studies have been done. Therefore, we think there is a need for more large-sample studies that can investigate the across- and within-country variation. However, the overview study by Johnson and Mislin is the best reference point we have at the moment. We have included a critical comment regarding this (see lines 734-735 of the revised version).

Reviewer #2: This manuscript touches off a very interesting topic and presents very powerful and rich data. The arguments and the conceptual model tested seem meaningful and thoughtful. However, there is huge room for improvement. On the one hand, I would suggest the authors to reduce as much as possible the main text and analyses in order to help the reader focus on the key elements. Currently, the paper seems unnecessarily long and complex to me. On the other hand, there are many typos, unorthodox expressions, and unclear parts, which make reading not to flow easily. I spot some of them below but my recommendation is to exert more effort and care in writing (which is not bad overall) and in structuring the flow of arguments. Below, I list more specific comments in (approx.) order of appearance in the text.

Response: We greatly appreciate the reviewer’s positive feedback on the merit of the paper, and we have done our best to address the critical and specific comments of the reviewer as described below. We are more uncertain about how to shorten or simplify the findings. We hope our clarifications and corrections have made it more readable. The paper has gone through professional editing.

1. Abstract (and elsewhere): “Generalized” trust typically refers to unaffiliated individuals, not to outgroups (who would also be “particularized”). See, e.g., Brewer, M. B. (1999). The psychology of prejudice: Ingroup love or outgroup hate?. Journal of social issues, 55, 429-444. Please check throughout the manuscript and justify the use of language.

Response: Thank you for this very useful reference! After careful reading of it, we have not changed our mind about our definition of outgroups as providing information about “generalized” social preferences and trust. Unknown outgroups and group members are unaffiliated to the within-group individuals who only have very limited knowledge of the outgroup members they play the game with. They know only that the outgroup members come from the same district (woreda) and belong to another youth business group. This means that they belong to the same ethnic group and we therefore do not expect any outgroup hostility due to outgroup members belonging to another ethnic group (considering the recent ethnic divides in Ethiopia). The outgroup characterization was chosen to create expectations that the outgroup members are in a similar situation in terms of wealth and opportunities (given that youth group members are resource-poor) such that responses should not be driven by expected relative wealth or opportunity differences. We therefore expect the difference in ingroup versus outgroup preferences to be due to the difference in social distance and social relations established within the closely knit youth business groups that on average had about 20 members who interacted frequently in person (they have a joint part-time business) and who have established internal rules of conduct and joint responsibilities (bylaws)). We therefore do not think the issues raised in the paper by Brewer (1999) on outgroup prejudice and discrimination driven by hate or antagonism are relevant in our study the way we framed the experiments. Where one draws the line between “ingroup” and “outgroup” depends on the purpose of the study but we have chosen to do it based on whether individuals have any direct personal relations or not and we have very clearly specified real world ingroups with only known members. Brewer (1999) defines ingroups as “bounded communities of mutual trust and obligation that delimit mutual interdependence and cooperation”. She also emphasizes an important aspect of this mutual trust as depersonalized (Brewer 1981) and extended to any member of the ingroup whether personally related or not (relies on group norm or expectations?). We acknowledge there is a continuum between clearly specified groups where members know each other well and larger social groups where most members do not have any personal relations, only cultural and ethnic origins in common. This contrasts with outgroups with different cultural and ethnic background. 

2. Abstract: please check “Our study has used incentivized”, better just “used”?

Response: Corrected (see line 27 of the revised version).

3. Line 39-40: “The pressures… is increasing” please check.

Response: Corrected to “The pressures… are increasing… (See lines 39-40 of the revised version).

4. Line 83: I would suggest referring to “outcome-based social preferences” or “distributional (social) preferences” to distinguish from reciprocity-based and type-based social preferences. See, e.g., Fehr, E., & Schmidt, K. M. (2006). The economics of fairness, reciprocity and altruism–experimental evidence and new theories. Handbook of the economics of giving, altruism and reciprocity, 1, 615-691.

Response: We appreciate this reference suggestion and now use this reference in the paper (See lines 88-93 of the revised version). Unlike this paper, however, we do not create specific utility functions that imply a choice between these three categories of models. We just identify social preference types but combine also with a reciprocity norm. We do not rule out interactions between preference types or intention-based preferences but also do not investigate such sequential interactions in detail. Such mechanisms may very well be part of the build-up of ingroup trust and may explain some of the variation we find in ingroup trust relative to outgroup trust. While F&S (2006) only present utility models based on behavior in games and without linking this to the real world, we use a simpler categorization of social preference types and our contribution is to link these to real world groups. Much of the study of group effects in experimental studies has been in “minimalist group” settings where the groups have been created in the game setting. This has advantages for certain purposes (internal validity and establishment of causality for certain randomized effects) but has limited external validity. 

5. Figure 1 is a bit messy. I encourage the authors to think carefully how to improve visualization. It also adds economic preferences on top of social preferences but there was not previous mention to economic preferences. It is also unclear why individual preferences do not impact on group performance. In addition, it is unclear why economic preferences are not labeled as “risk preferences”.

Response: We have improved the visualization of Figure 1. Individual (social and economic) preferences influence group decisions, collective action, and group activities as well as the individual decisions in the games. In Figure 1, we consider selfish preferences (maximizing personal gain) as well as risk preferences as economic preferences. In the trust game there is a trade-off between maximizing individual return and minimizing risk to investment in the trust game. Group performance is a result of group activity and where social norms, trust and trustworthiness are the intermediate outcomes that are influenced by individual preferences and the interactions among group members in their collective action activities. We therefore argue that there is no direct link from the individual preferences to the group performance (which is an outcome at group level and is a result of joint efforts). The group activity is a part-time activity for members and a complementary source of livelihood. In the revised paper we have clarified this. Individual perceptions of group performance are used as indicators of group performance. 

6. Figure 2a: it is unclear what the error bars represent. Standard errors? Confidence intervals? Please clarify.

Response: The bars represent 95% confidence intervals and we stated this in the legend in the revised version. 

7. Figure 2b and elsewhere: “net trust gain” is an unorthodox label for the standard “ingroup favoritism in trust”, “ingroup bias in trust”, or “intergroup bias in trust”. The problem is that the label is misleading (why “gain”?). Also, please clarify why the kernel density plots use apparently different bandwidth specifications.

Response: We recognize that “net trust gain” is an unorthodox term. The logic behind the term is that there is a gain in the level of trust within closely knit groups as compared to generalized trust at the broader level. We estimated the degree of such gain using average outgroup trust of the same group members as the benchmark. We could have called it ingroup bias in trust as well, but “bias” and “favoritism” may also trigger some false ideas about the measure being “biased” in a statistical sense or being inappropriate. The higher ingroup trust level may be very well founded and based on the repeated interactions within groups and thereby being a good group performance indicator. We therefore still prefer our choice of term for it even though it is new. We also rephrase it as ingroup trust gain (relative to outgroup trust). It is a form of social capital that the group has built through its interactions. 

On the kernel density bandwidths, we are not sure which graphs the reviewer has in mind. Stata typically sets the bandwidth automatically as follows: “bwidth (#) specifies the half-width of the kernel, the width of the density window around each point. If bwidth() is not specified, the "optimal" width is calculated and used; see [R] kdensity. The optimal width is the width that would minimize the mean integrated squared error if the data were Gaussian and a Gaussian kernel were used, so it is not optimal in any global sense. In fact, for multimodal and highly skewed densities, this width is usually too wide and over smooths the density (Silverman 1986)”. We typically may adjust the bandwidth if the visual picture is too smooth or too noisy.

8. Line 263: “The order of the games was the same for all respondents for practical reasons”. Please list more clearly this as a limitation of the study.

Response: We have corrected and stated that this is a limitation of the study (See lines 298-301 of the revised version). We think that these games are so simple that there is not much learning taking place in the sequence of the games. We still cannot rule out an order bias although we think it is likely to be low. However, such a potential order bias applies equally to the whole sample and even if it is significant, we do not think it is important for the overall results. We would be more concerned about this if the games were more cognitively demanding and requiring substantial numeracy skills. Given the complexity of implementing high quality field experiments, we decided that this simplification would help minimize implementation errors which could be more of a problem than the order bias we may have. 

9. Table B: although it is not impossible, it seems strange that age is positively related to both altruistic and spiteful preferences (ingroup) which are exactly the opposite preferences. Please mention about this result and why it is not strange.

Response: It is important to note that the reference category in the econometric analysis is the residual group of members that were not assigned to any of the four main types. While a further sub-categorization of this residual group into weak altruist and weak egalitarian could have been possible, we chose to lump these together as the reference category that does not signal any strong social preference or strong selfish preference. We now explain this in relation to the categorization of social preference in Table A of S2 Appendix. The signs for age related to altruistic and spiteful preferences may be related to the relatively small shares they represent of the full sample.

10. Figure 3 (results on ingroup favoritism in social preferences): these results might be compared to those of Espín et al. (2019) where a similar analysis was performed. Espín, A. M., Correa, M., & Ruiz-Villaverde, A. (2019). Patience predicts cooperative synergy: The roles of ingroup bias and reciprocity. Journal of Behavioral and Experimental Economics, 83, 101465.

Response: Thank you for this valuable and interesting reference! While its primary focus is on patience, a dimension we have not included in our study, the social preference and ingroup and outgroup assessments are relevant. However, the context is very different as it is a classroom study with university students where intergroup competition is created as a treatment. We do not think such intergroup competition is important for the performance in our real-world groups. Rather it may be the assessment of group performance by local government administrations and by the group’s own member assessments (satisfaction) that are important in our case. The hypothetical social preference games in the paper are similar to our incentivized games. However, the large differences in focus (patience and competition between groups) context and framing make us reluctant to make direct comparisons of the results but we may have missed an important point here. The paper has triggered our curiosity about the relationship between patience and social preferences and we have time preference data for the main sample but will have to pursue this in another paper where this reference will be valuable for us. 

11. Line 307: “More altruistic preferences may also become “epidemic” within groups.” I do not understand what “epidemic” means in this context.

Response: What we had in mind is that conditional reciprocity may have such a within-group effect that altruists generate more altruists within the group. E.g. Figure 3a shows that the maximum share of altruists in the outgroup context is 0.4 while it is 0.8 in the ingroup context. We have added this explanation (see lines 349-350 of the revised version).

12. Line 310: please check “are rare in both ingroup context but”.

Response: Corrected. It should read “…are rare in both ingroup and outgroup contexts but…” (see line 352 of the revised version).

13. Line 326: “group average net trust gain (the difference between average outgroup and ingroup trust)” Shouldn’t be the opposite? i.e., difference between ingroup and outgroup.

Response: Corrected to “The group average net ingroup trust gain (the difference between average ingroup and outgroup trust) is also varying substantially but is positive for all groups (see lines 369-370 of the revised version)”.

14. Line 371: it is unclear what the “Z variable” refers to.

Response: Corrected. We forgot to change Z when changing the notation in the equations. It should be Et and represents community (tabia) dummies to control for community fixed effects (See line 415 of the revised version).

15. Line 528: please check “the results for generalized trust and trustworthiness the results”.

Response: Corrected and it should read as “the results for generalized trust and trustworthiness… (see lines 577-579 of the revised version)”.

16. Line 565 and elsewhere: significance should not be qualified as “highly” (or any other qualification). Line 582: “less significant” is not a correct expression either. Please check throughout the manuscript.

Response: Corrected as per the comment (See line 616 and elsewhere).

17. Table 5 and elsewhere: “System of equations model” does not seem the most standard label for “structural equation model”.

Response: Corrected (see Table 6 and elsewhere).

18. Table 7: it seems that (at least in my pdf) the variable “egalitarian, dummy” is missing. Also, here and elsewhere, the “norm to reciprocate” variable seems defined in the wrong way (line 368). Please define it as a positive function of reciprocation norm, not as a negative function.

Response: The missing variable “egalitarian, dummy” in Table 7 (Now Table 8 in the revised version) is included. We have also reversed the order of response categories for the reciprocity norm variables such that a stronger norm to reciprocate is associated with a higher rank number. This results in a reverse of the sign for these variables where they are RHS variables and for the other RHS variables in the models where these variables are dependent variables (Tables 6 and 8 in the revised version).

19. Line 665: “Thus, generalized norms of reciprocity and trust were less exogenous than we had hypothesized.” I do not understand this statement (Figure 1 shows a causal effect of social preferences on trust and social norms). Please clarify.

Response: It means that the strength of the stated social norm is a function of social preferences and their within-group distribution rather than an exogenous variable. But we agree that “what we have hypothesized” is misleading, given Figure 1. We have corrected to “in line with the hypotheses in Fig 1”(See lines 714-715 of the revised version). 

20. Lines 669-678: the aggregate % of each type might be meaningfully comparable to those of Corgnet et al. (2015) for example (at least an adult sample)? Corgnet, B., Espín, A. M., & Hernán-González, R. (2015). The cognitive basis of social behavior: cognitive reflection overrides antisocial but not always prosocial motives. Frontiers in Behavioral Neuroscience, 9, 287.

Response: Thank you for this very interesting reference, we reviewed it (See line 300 of the revised version and reference list number 44). We also included some cognitive tests (numeracy skills) in our study but have not yet investigated how they relate to the social preferences. They were intended for the study of time and risk preferences where we had clearer theoretical ideas about their relevance. However, we did not use the Cognitive Reflection Test in our study areas. We refer to the paper related to possible reflection effects associated with the fixed order of the sequence of our social preference games. The later games may be associated with more time to reflect than the first ones. However, all are treated equally in this respect. 

21. Line 698: “The net trust gain was negatively correlated with outgroup trust.” Of course, since “net trust gain” is defined as the difference between ingroup and outgroup trust. This seems misleading.

Response: We do not understand why this is misleading? The higher level you start from the less incremental improvement is feasible given that these are measures within the 0-1 range. We have rephrased it to “net ingroup trust gain” to make the term clearer (see line 749 and elsewhere in the revised version). 

22. As a general comment, how can we be sure that the causal relationships are not different from those proposed in Figure 1? Is there any way to compare against alternative specifications?

Response: This would require alternative (competing) theories. We only tested alternative statistical models (S2 Appendix) which gave remarkably similar results. There may potentially be many complementary and possibly competing theories that could be tested. However, this goes beyond the scope of this paper which already covers a lot! E.g. we have rich additional data on time and risk preferences of the group members and more work on these data are in the pipeline. The ongoing civil war in the study areas makes it very uncertain what has happened to these youth after the war broke out. Our study may therefore also potentially become a valuable baseline for studies of the war impacts on the youth.

---

## [Decision Letter · Decision Letter 1]

6 Sep 2021

Preferences, trust, and performance in youth business groups

PONE-D-21-15257R1

Dear Dr. Gelaye,

We’re pleased to inform you that your manuscript has been judged scientifically suitable for publication and will be formally accepted for publication once it meets all outstanding technical requirements.

Kind regards,

Pablo Brañas-Garza, PhD Economics

Academic Editor

PLOS ONE

Additional Editor Comments (optional):

Reviewers' comments:

Reviewer's Responses to Questions

**Comments to the Author**

1. If the authors have adequately addressed your comments raised in a previous round of review and you feel that this manuscript is now acceptable for publication, you may indicate that here to bypass the “Comments to the Author” section, enter your conflict of interest statement in the “Confidential to Editor” section, and submit your "Accept" recommendation.

Reviewer #1: All comments have been addressed

Reviewer #2: All comments have been addressed

2. Is the manuscript technically sound, and do the data support the conclusions?

Reviewer #1: Yes

Reviewer #2: Yes

3. Has the statistical analysis been performed appropriately and rigorously? 

Reviewer #1: Yes

Reviewer #2: Yes

4. Have the authors made all data underlying the findings in their manuscript fully available?

Reviewer #1: Yes

Reviewer #2: Yes

5. Is the manuscript presented in an intelligible fashion and written in standard English?

Reviewer #1: Yes

Reviewer #2: Yes

6. Review Comments to the Author

Reviewer #1: Please, review this title: Table 1. Demographic characteristics of sample youth group members by district.

Reviewer #2: (No Response)

7. PLOS authors have the option to publish the peer review history of their article (what does this mean?). If published, this will include your full peer review and any attached files.

Reviewer #1: No

Reviewer #2: No

---

## [Editor Report · Acceptance letter]

9 Sep 2021

PONE-D-21-15257R1 

Preferences, trust, and performance in youth business groups 

Dear Dr. Tilahun:

I'm pleased to inform you that your manuscript has been deemed suitable for publication in PLOS ONE. Congratulations! Your manuscript is now with our production department. 

Kind regards, 

on behalf of

Dr Pablo Brañas-Garza 

Academic Editor

PLOS ONE